# Mechanisms for the emergence of Gaussian correlations

Marek Gluza[1,2]⋆, Thomas Schweigler[3,4], Mohammadamin Tajik[3], João Sabino[3,5,6], Federica Cataldini[3], Frederik S. Møller[3], Si-Cong Ji[3], Bernhard Rauer[3,7], Jörg Schmiedmayer[3], Jens Eisert[1,8] and Spyros Sotiriadis[1,9]

**1** Dahlem Center for Complex Quantum Systems,
Freie Universität Berlin, 14195 Berlin, Germany
**2** School of Physical and Mathematical Sciences, Nanyang Technological University, 21
Nanyang Link, 637371 Singapore, Republic of Singapore
**3** Vienna Center for Quantum Science and Technology, Atominstitut,
TU Wien, 1020 Vienna, Austria
**4** JILA, University of Colorado, Boulder, Colorado 80309-0440, USA
**5** Instituto de Telecomunicações, Physics of Information and Quantum Technologies Group,
Av. Rovisco Pais 1, 1049-001, Lisbon, Portugal
**6** Instituto Superior Técnico, Universidade de Lisboa,
Av. Rovisco Pais 1, 1049-001, Lisbon, Portugal
**7** Laboratoire Kastler Brossel, Ecole Normale Supérieure,
24 rue Lhomond, F-75231 Paris, France
**8** Helmholtz Center Berlin, 14109 Berlin, Germany
**9** Department of Physics, University of Ljubljana, 1000 Ljubljana, Slovenia

⋆ marekludwik.gluza@ntu.edu.sg

## Abstract

We comprehensively investigate two distinct mechanisms leading to memory loss of non-Gaussian correlations after switching off the interactions in an isolated quantum system undergoing out-of-equilibrium dynamics. The first mechanism is based on spatial scrambling and results in the emergence of locally Gaussian steady states in large systems evolving over long times. The second mechanism, characterized as 'canonical transmutation', is based on the mixing of a pair of canonically conjugate fields, one of which initially exhibits non-Gaussian fluctuations while the other is Gaussian and dominates the dynamics, resulting in the emergence of relative Gaussianity even at finite system sizes and times. We evaluate signatures of the occurrence of the two candidate mechanisms in a recent experiment that has observed Gaussification in an atom-chip controlled ultracold gas and elucidate evidence that it is canonical transmutation rather than spatial scrambling that is responsible for Gaussification in the experiment. Both mechanisms are shown to share the common feature that the Gaussian correlations revealed dynamically by the quench are already present though practically inaccessible at the initial time. On the way, we present novel observations based on the experimental data, demonstrating clustering of equilibrium correlations, analyzing the dynamics of full counting statistics, and utilizing tomographic reconstructions of quantum field states. Our work aims at providing an accessible presentation of the potential of atom-chip experiments to explore fundamental aspects of quantum field theories in quantum simulations.



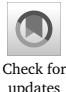

# Contents

# 1 Introduction

By appropriately choosing the effective degrees of freedom, it is frequently possible to capture complex collective behavior of an interacting quantum many-body system using a simple Gaussian model. There is an abundance of exact analytical tools for computing physical properties using such Gaussian effective descriptions. Most crucially, due to Wick's theorem the second moments are decisive for higher order correlation functions which do not yield further information. Thanks to such concise features in sync with a high predictive power, it is fair to say that Gaussian models are the bread and butter of many physicists, independent of their focus, be it experimental or theoretical.

The study of far from equilibrium quantum dynamics, however, hints at a fundamental question concerning the applicability of such Gaussian models. Whenever many-body interactions are sufficiently strong so that they can no longer be meaningfully neglected, their properties become imprinted onto the correlations in the system. But what will happen if these interactions are suddenly removed? After such a quench protocol – a ubiquitous one actually considered in a quite large body of literature [1–4] – the ensuing time evolution will be Gaussian and governed by a non-interacting quadratic Hamiltonian. Will the state over time lose memory of those initial interactions? And if so, in what precise way?

This is quite a complex physics question in that immediately after the interaction quench, there is an apparent discrepancy between the state (which is non-Gaussian) and the (Gaussian) Hamiltonian closed system dynamics. For quantum states with Gaussian correlations, all correlations can be determined from first and second moments of canonical coordinates using Wick's theorem. A natural expectation may be that for long times, the discrepancy will become less stark, eventually the state becoming approximately Gaussian, in accordance with the Hamiltonian governing the dynamics. A complication arises, however, because quadratic Hamiltonians which give rise to Gaussian dynamics feature a large number of conserved charges, which may be relevant in a given quench scenario leading to intricate memory effects and hence complicating the task of understanding the dynamics following an interaction quench.

More specifically, conserved charges may enable a physical realization of the persistence of the discrepancy between the initial non-Gaussian state and the quadratic Hamiltonian. Moreover, even independently of the presence of conserved charges, the nature of the dynamics may influence the timescale for the decay of the discrepancy between the non-Gaussian initial state and the Gaussian quench dynamics. Thanks to the Gaussian character of the quench dynamics, such memory loss of the signatures of interactions present prior to the quench can be characterized theoretically for certain scenarios [5–7]. Thus, it is natural to ask whether the existing theoretical results can be used to explain the decay of non-Gaussianity of a state whenever it appears experimentally. As a wider perspective, the question that is on the desk is a particular reading of the question of the emergence of a *generalized Gibbs ensemble* [5,8–12]. Understanding the mechanism and conditions for the emergence of Gaussian correlations is a good opportunity to understand more generally the nature of equilibration in isolated quantum systems, since Gaussian dynamics is an exceptionally convenient case for an in-depth theory-experiment comparison. This is due to two main reasons. On the theoretical side, exact analytical methods are available for the study of not only the final steady state, but also for the full dynamics of correlations of any order and for arbitrary initial states [5–7, 13–21]. From an experimental perspective, the possibility to measure and characterize the factorization properties of higher-order correlations first achieved in Refs. [22, 23] offers a practically complete description of quantum states and their dynamics.

The main aim of this work is to elaborate on both theoretical and experimental viewpoints on the question how quantum states become Gaussian over time in the context of the exper-

imental findings recently presented in Ref. [24]. In that experimental work, non-Gaussian initial states have been prepared through the coupling between two adjacent one-dimensional ultra-cold gases, and after a fast decoupling that corresponds effectively to an interaction switch-off quench, a decay and subsequently a revival of non-Gaussianity has been observed. In the remainder of this introductory section, we will present the essential ideas behind two distinct theoretical mechanisms which yield memory loss of non-Gaussianity in the system. The first of the two mechanisms is an instance of those studied in earlier theoretical works, while the second one has been introduced in Ref. [24]. We will then lay out in more detail the precise experimental observations presented in Ref. [24]. In Section 2 the two mechanisms, which can be considered as candidates to explain the experimental observations, will be presented in detail. Each of the two mechanisms relies on the presence of different properties of the initial states and of the dynamics. The characteristics of the initial states in the experiment relevant to corroborate the two candidate mechanisms will be presented in Section 3 and those related to dynamical aspects of the experimental system will be presented in Section 4. Further observations based on the experimental data are discussed in Section 5. By combining the conclusions of all previous sections we finally evaluate the role of each of the two mechanisms in Subsection 5.3 and conclude that the one that is mainly responsible for the emergence of Gaussianity in the experiment is the second. At the end of the manuscript we provide conclusions and outlook in Section 6.

Our results are to a large extent based on the use of a specific reading of quantum field tomography. In the present context, this term refers to an indirect recovery of all second moments of quadratures [25] from an ensemble of identically prepared quantum states, exploiting suitable time evolution under non-interacting Hamiltonians. It is a recovery method reminiscent of full quantum state recovery based on data [26]. After all, a quantum state of a quantum many-body system can be characterized by the collection of all moments of observables [22]. It is important to stress that the experimental prescription allows for the reconstruction of higher moments of quadratures as well [22], an insight that allows us to monitor the process of Gaussification in time. The tomographic method allows us to verify that the conditions of the second Gaussification mechanism are satisfied to a sufficient degree in the experiment by direct analysis of the experimental data (Subsection 3.3.2). Preliminary evidence for the validity of these requirements has been presented in Ref. [24] using simulations based on the classical fields approximation [27].

In the course of our analysis, we derive further results of independent interest, including an analysis of the scaling of correlations in the initial equilibrium states (Subsection 3.1), a demonstration of light-cone propagation of correlations (Subsection 4.1) and a study of the dynamics of full counting statistics (Subsection 5.1). Lastly, we summarize the theoretical description of the experimental system by means of TLL theory in the Appendix, justifying why deviations from this theoretical model are negligible based directly on experimental observations. Overall, our work provides a detailed comparison between theory and experiment that aims to contribute to our understanding of the physics of one-dimensional coupled atomic condensates and their dynamics.

## 1.1 Two mechanisms for the emergence of Gaussian correlations

We aim to precisely understand what happens when a system lies initially in a non-Gaussian state and subsequently evolves under Gaussian dynamics. Any process wherein non-Gaussian correlations in an isolated quantum system decay over time rendering the resulting state effectively Gaussian will be called in this work 'Gaussification'.

In nature, we often find that a description of the system by means of statistical mechanics is instructive and accurate. For this to be true when starting out of equilibrium, some sort of scrambling of information encoded in the initial state must occur during the dynamics in one

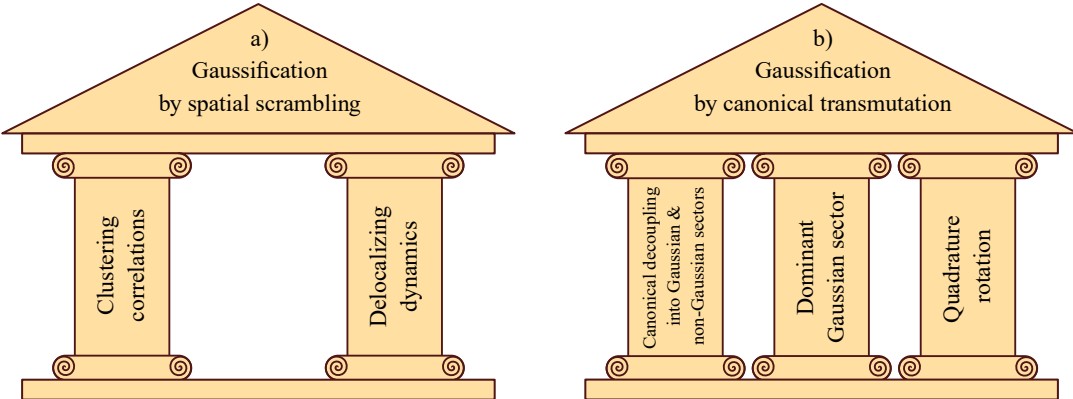

Figure 1: After an interaction quench in an isolated system, the memory of interactions as encoded in the initial state through non-Gaussian correlations can gradually decay by means of two distinct mechanisms (broadly introduced in Subsection 1.1 and discussed in detail in Section 2): **a)** Gaussification by spatial scrambling (Subsection 2.1) rests on two pillars. The first is a property of the initial state to yield independent results in measurements between distant points, which is briefly called clustering correlations. Its validity in the experiment is discussed in Subsection 3.1. The second pillar is a property of the dynamics to make initially local fluctuations spread over large portions of the system during the time evolution, and its validity is tested in Subsections 4.1 and 4.2. **b)** Gaussification by canonical transmutation is the second mechanism discussed in this work (Subsection 2.2). It relies on the initial state having at least one Gaussian canonical sector (Subsection 3.2) that dominates the other sector which is non-Gaussian (Subsection 3.3). In this mechanism, quadrature rotation during the dynamics results in the initially non-Gaussian quadrature turning into Gaussian. In coordinate space, this gives rise to dephasing and rephasing dynamics which we will broadly refer to as canonical transmutation dynamics.

way or the other. We expect this to occur generically, under interacting or weakly interacting dynamics, in accordance with classical intuition built around Boltzmann's *H*-theorem and phenomenology surrounding kinetic equations. Such scrambling of information and memory loss effects in isolated quantum systems can be induced by simple Gaussian dynamics. The induced memory loss can be sufficient to give rise to an agreement between local reductions of the unitarily evolved density matrix and the respective marginals of a Gaussian steady state [5–7, 14, 16–19].

One such process of Gaussification – as we will argue – occurs in conjunction with a notion of what we call spatial scrambling. Intuitively, in a large system as the elapsed time becomes long, local observables depend on larger and larger amounts of incoherent initial information originating from distant points. If distant points have initially been only weakly correlated then this results in an elimination of non-Gaussian features in correlation functions. The unitary Gaussian dynamics implements it in a way that is arguably in reminiscence of classical central limit theorems [7, 14]. In fact, a precise connection to mathematical proofs of Lindeberg central limit theorems on the level of characteristic functions can be established [14].

The mechanism of Gaussification by *spatial scrambling* rests on two essential physical ingredients. Firstly, the initial correlations of the effective fields in terms of which the dynamics is Gaussian must satisfy the condition of clustering, i.e., their correlations between distant points must factorize as for independent variables, or, equivalently, their connected correlations must decay with distance. The real-space distance scale for this is commonly set

by the correlation length. Weaker conditions (like algebraic instead of exponential clustering or sub-extensivity of the initial fluctuations of macroscopic conserved quantities [21]) can also play the same role, with the main physical condition on the initial state being that it has local characteristics like the equilibrium states of systems with local interactions. Such conditions are quite ubiquitously valid in nature. This is important because statistical mechanics has wide applicability so the prerequisites for its emergence in isolated systems should be broadly fulfilled. Secondly, the dynamics must induce delocalization, i.e., an initially localized fluctuation of the effective field must spread with time, not remain close to its original position or just rigidly move through the system during the evolution. In Gaussian dynamics such a behavior is quite typical and can be linked to properties of the energy dispersion relation, specifically its non-linearity. These two broadly valid conditions in essence imply a Gaussification process [5, 7, 16–20]. Fig. 1 provides an illustration of these 'pillars' and summary for reference during the reading of this work, including information on which section discusses each of the 'pillars'.

While the pillars on which Gaussification by spatial scrambling rests are quite general, their applicability cannot be taken for granted. A particularly interesting case where this is not given a priori occurs when the collective fields providing the Gaussian effective description of the system's dynamics are non-locally related to the physical local degrees of freedom (e.g. particles), or if the spectrum of the effective fields is to a good approximation linear. A non-local relation between local physical fields and the collective fields of the effective description might mean that the initial clustering condition is not guaranteed for the latter, even if valid for the former: indeed, while clustering is a typical physical requirement for local fields and observables, it does not have to be satisfied by non-local collective fields. Moreover, when the dispersion relation of the collective field excitations is linear, then, even if these fields are genuinely local, the dynamical delocalization condition is broken because the time evolution does not induce spreading of initially localized wave-packets which travel instead pinned at two moving points. Both these aspects in question are directly motivated by the experimental study of a quench of the effective interaction between phononic excitations in coupled one-dimensional ultra-cold gases.

Specifically, in Ref. [24], a decay of non-Gaussian correlations in time has been observed and a novel mechanism for its explanation has been proposed, a mechanism which we will call in this work Gaussification by *canonical transmutation*. This mechanism is at play when the initial state capturing the correlations of two canonically conjugate fields yields Gaussian correlations for one canonical variable and non-Gaussian for the other. A simple harmonic phase-space rotation of the system's eigen-modes after switching off the interaction may then lead to the decay of non-Gaussianity due to dilution of the non-Gaussian into the Gaussian component if the latter dominates in the mixing process. Such a change in the internal make-up of an object changing drastically its overall properties seems to agree with the general meaning of the word 'transmutation', so for lack of a better term we will consistently employ it in this work to make clear which of the two mechanisms we are referring to. As we will discuss in detail later, this mechanism rests on three 'pillars' as summarized and illustrated in Fig. 1, and the individual ingredients will be discussed one-by-one in the following. Again, we speak of pillars in the sense that they seem to be necessary for the effect as the absence of each one of them breaks down the mechanism and leads to preservation of the memory of non-Gaussian correlations.

## 1.2 Experimental observation of the decay and recurrence of non-Gaussian correlations

Having introduced the two theoretical mechanisms of Gaussification, our goal is to investigate whether they can explain the observed decay of non-Gaussian correlations in the experiment

of Ref. [24]. Let us look into the context and findings of this experiment in more detail. We will give a description of the experiment and of the analysis of measured data and summarize the main results.

In principle, an overall intuition regarding the main experimental observation can be obtained based on a purely statistical consideration: The experiment yields outcomes which differ from one experimental realization to another. The outcomes of measurements are treated as instances of random variables so that from this sample estimates of statistical moments of these variables can be extracted. If the fourth and higher order cumulants are negligible then we speak of a Gaussian state; in the opposite case we have a non-Gaussian state.

The experimentally measured variables (phase and density of the atomic gas) constitute collective degrees of freedom in an effective description of the actual many-body system. By varying a parameter of the system and measuring the statistical moments when the system is at equilibrium, we can study how the parameter controls the non-Gaussianity of equilibrium states in the effective description. From the observation that the size of non-Gaussianity depends on the parameter under consideration, just from statistical considerations we conclude that a non-linear *effective interaction* must be present and can be controlled by the above parameter.

In addition, by rapidly changing the parameter and measuring the non-Gaussianity at various subsequent times, we can study how this changes over time. The experiment has displayed a decay of non-Gaussianity as a function of the time elapsed after switching the parameter from the non-Gaussian to the Gaussian regime. This may sound quite reasonable intuitively: the system dynamically adapts to the change in the external parameter relaxing to the corresponding equilibrium state. However, one aspect of the dynamics studied in Ref. [24] evades a simple intuitive interpretation. How should one interpret the fact that after a monotonous decay of non-Gaussianity the experiment shows a revival of the non-Gaussian correlations at a later time? To answer this question we first need to know a crucial piece of information about the dynamics. To a very good approximation the system evolves as a *closed* one: interaction with the environment is strongly suppressed over the time scale of the experiment, so that the system is practically isolated. Under such settings the *information* about the system being initially non-Gaussian is never lost but gets first hidden from view, resulting in a Gaussian state at intermediate times, and is subsequently retrieved again at the time of the revival of non-Gaussianity. Hence, given that the dynamics is essentially unitary, the information about the initial state has never left the system and has not been irreversibly scrambled. The emergence of a Gaussian equilibrium-like state is now somewhat less obvious and the mechanism leading to it deserves investigation.

We will now proceed to explaining how this physical picture has been realized in the experiment. A description of the experiment and data analysis is illustrated in Fig. 2. The system considered consists of two parallel and adjacent one-dimensional quasi-condensates of a few thousand rubidium atoms trapped, cooled and controlled by means of an *atom-chip* setup (see Fig. 2.a). The experimental measurements are interference pictures generated when the two-component gas is released from the trap and let to expand freely. From the interferometric measurements we extract phase profiles $\phi(z, t)$ corresponding to the relative phase between the two components of the gas as a function of the longitudinal dimension $z$ at the time of the release $t$. By repeating the experiment many (several hundreds of) times we obtain an ensemble of phase profiles from which we can derive statistical measures of the phase field distribution: correlation functions between different points, moments, cumulants, or the full distribution function of the phase.

The relative phase profiles play the role of the fundamental random variables in our statistical consideration. In practice, a phase profile is a vector whose entries correspond to the

SciPost Phys. **12**, 113 (2022)



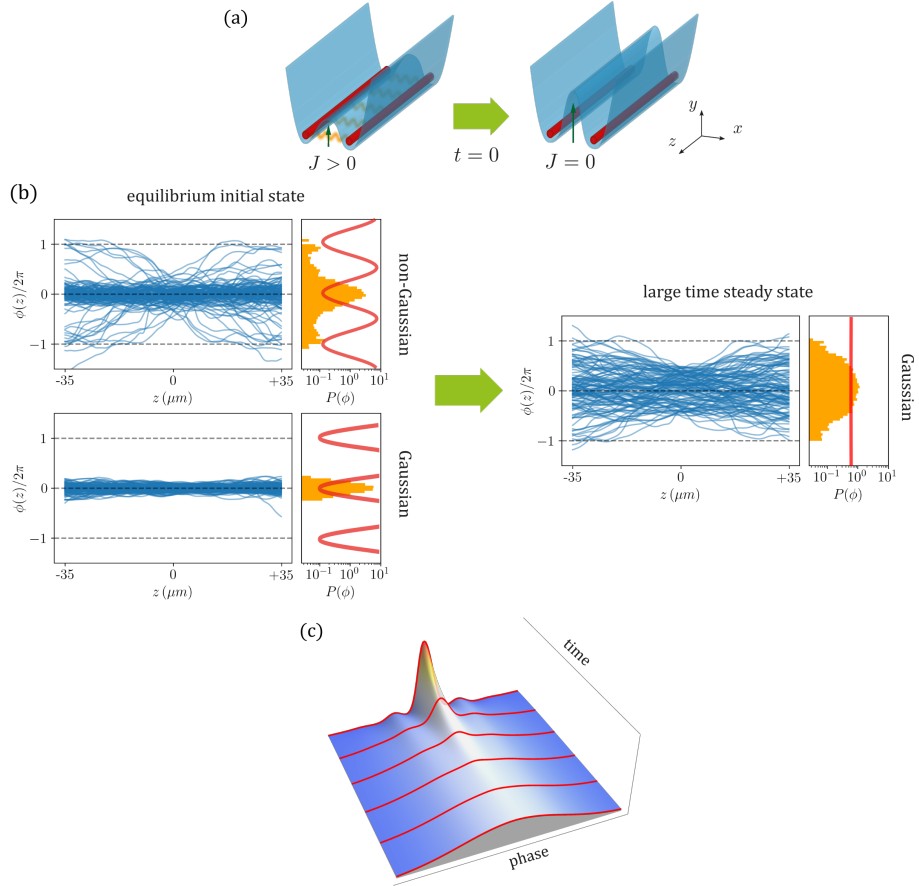

Figure 2: a) Illustration of the experimental setup: A gas of ultra-cold atoms confined into a one-dimensional trap can be split in two adjacent parallel traps controlled by an atom chip. By controlling the barrier height of the transverse double well potential which plays the role of a Josephson junction, one can engineer a many-body interaction of the cosine type. Ramping up the barrier height amounts to switching the interaction off. b) Preparing the gas at an equilibrium state and releasing it from the trap, the interference between the two atomic clouds reveals information about the spatial profile $\phi(z,t)$ of their relative phase at the time of the release. Repeating the experiment many times results in an ensemble of phase profiles from which one can extract the probability distribution of various physical observables derivable from the phase field. In the presence of interaction and at sufficiently high temperature, the distribution of the relative phase is highly non-Gaussian, exhibiting a central peak at $\phi = 0$ and accompanied by side peaks at $\phi = \pm 2\pi$ with a lower height. This pattern clearly indicates the presence of solitons in the phase profiles and the concentration of the phase at the minima of the cosine potential. If the coupling $J$ is very large, the interaction potential is approximately a steep parabola and phase fluctuations are Gaussian. Changing the interaction strength $J$ from a nonzero value to zero and monitoring the subsequent time evolution of the phase profiles, we observe the transition from an initial non-Gaussian to an emergent Gaussian distribution. The plots show typical examples of equal-time ensembles of phase profiles and the corresponding histograms of phase values in all points $z$, as measured in the experiment. The three cases correspond to two different types of initial states (at intermediate or large $J$) and a typical state that emerges after quenching to $J = 0$. (c) Illustration of the time evolution of the phase distribution.

estimate of the relative phase between the two quasi-condensates at the position corresponding to a given pixel of the read-out camera [22, 28–33]. The phase observable is modelled by a bosonic field $\hat{\varphi}(.)$ ranging over the extension of the one-dimensional quasi-condensate, whose excitations have the physical interpretation of phonons.

The two quasi-condensates are trapped in a transversal double well as illustrated in Fig. 2.a with a barrier that controls the tunneling strength $J$ between the two wells. This tunnel coupling is quite well characterized: For intermediate values of $J$ (relative to the temperature) it leads to an effective many-body interaction of the phonons and hence the state of the system that is prepared at equilibrium at such couplings is non-Gaussian. This interaction term is known to agree well with a *sine-Gordon potential*, as predicted in Refs. [34, 35] and verified experimentally in Ref. [22]. The latter work has also shown that in the limit cases of small or large $J$ a Gaussian distribution is obtained. One simple way to assess the non-Gaussianity of the system's state is to disregard the spatial variation of the phase profiles in the $z$-direction and consider a cumulative histogram. The question is then how far the obtained distribution is from a Gaussian. Fig. 2.b illustrates two typical cases. In the case of intermediate $J$ discussed above the histogram exhibits long tails or even shoulder peaks that signify deviations from a regular Gaussian distribution. In the case of small or large $J$ on the contrary the distribution is plain Gaussian.

Ramping up the barrier height of the double-well between the two adjacent quasi-condensates, as illustrated in Fig. 2.a, reduces the tunneling of atoms between the two wells, hence reducing the $J$ parameter to zero. Doing this from the intermediate regime where interactions play initially a substantial role corresponds to switching off the effective interaction, which is the quench situation that we want to consider. The resulting dynamics of the histograms is illustrated in Fig. 2.c: An initial distribution with shoulders that constitute deviations from a Gaussian evolves so that the deviations decay over time.

Fig. 2 is at this stage an illustration of a subset of questions one can study experimentally using the atom-chip platform. We will next discuss a more space-resolved way of assessing the non-Gaussianity of the system using correlation functions that has been presented in Ref. [24]. Nonetheless, the dynamical behavior illustrated in Fig. 2.c will make an appearance towards the end of the manuscript in Subsection 5.1 where we will present how histograms of the type illustrated in Fig. 2.b vary in time.

The measurement outcomes of the phononic phase field $\hat{\varphi}(.)$ can be viewed as spatially resolved values of the relative phase between the two quasi-condensates. At this point, it should be noted that given that the relative phase is an angular variable its value at any point can only be measured with an ambiguity of a $2\pi n$ shift where $n$ is an integer. The measured phase profiles are derived by imposing the condition that the phase at some reference point is within the interval $[-\pi, +\pi)$ and then 'unwrapping' the phase profile so that values at any two neighboring points differ no more than $\pi$. The ensembles of phase profiles shown in Fig. 2.b are extracted in precisely this way, where the reference point is fixed to the middle. This procedure still involves however an arbitrary choice of an overall $2\pi n$ phase shift. To completely remove this ambiguity we can restrict our analysis to observables calculated strictly on phase differences between any point and some (arbitrarily chosen) reference point. For this reason, we define the observable phase difference with respect to the reference point $z_0$ as

$$\Delta\hat{\varphi}(z) := \hat{\varphi}(z) - \hat{\varphi}(z_0). \tag{1}$$

By repetition of the experiment and interferometric measurement under identical conditions, the full correlation function can be estimated as

$$\Phi(z_1, \dots z_n) = \left\langle \prod_{i=1}^{n} \Delta\hat{\varphi}(z_i) \right\rangle. \tag{2}$$

This is the statistical moment that we have defined above. Taking $n = 1$ gives just the average phase that typically vanishes due to effective symmetries of the physical processes involved in the state preparations. For $n = 2$, we obtain the second moments which allow us to parametrize a Gaussian distribution and to build higher-order correlations using Wick's theorem.

Noticing that correlation functions of odd order are negligible, we find that for $n = 4$, the *connected* correlation functions take the form

$$\Phi_{\text{con}}(z_1, \ldots z_4) = \Phi(z_1, \ldots z_4) - \Phi_{\text{Wick}}(z_1, \ldots z_4), \tag{3}$$

where we just subtract the standard *Wick* decomposition

$$\Phi_{\text{Wick}}(z_1, z_2, z_3, z_4) = \Phi(z_1, z_2)\Phi(z_3, z_4) + \Phi(z_1, z_3)\Phi(z_2, z_4) + \Phi(z_1, z_4)\Phi(z_2, z_3), \tag{4}$$

an expression in which the sum ranges over all possible pairings of indices. We see that such a 4-point connected function vanishes for all Gaussian distributions because then $\Phi(z_1, \ldots z_4) = \Phi_{\text{Wick}}(z_1, \ldots z_4)$ essentially by definition.

Such an analysis, i.e., extraction of various moments and evaluation of connected functions, depends only on the measured phase profiles and hence until now we did not indicate any time dependence. The time dependence is encoded in the quantum state at the time of the measurement from which the phase profiles are being sampled. That is, each time $t$ we measure, after quenching the tunneling parameter $J$, the system is described by some density matrix $\hat{\varrho}(t)$ and the expectation values $\langle \cdot \rangle$ in Eqs. (2-4) refer to this density matrix. When the system involves many degrees of freedom that interact with each other, then its state is quite complex and it is not practical to inquire about its entire quantum state $\hat{\varrho}(t)$. Instead, one should consider correlation functions and indeed these can be obtained experimentally and be used to characterize the system at different times $t$ [22, 23]. Accordingly, we will indicate correlation functions obtained at different times by writing explicitly their dependence on first the spatial variables and then the time variable, as $\Phi(z_1, \ldots, z_n, t)$.

We are now in a position to discuss the measure for the non-Gaussianity of the phase fluctuations based on correlation functions that has been presented in Refs. [22, 24]. It is given by

$$M^{(4)}(t) = \frac{S_{\text{con}}^{(4)}(t)}{S_{\text{full}}^{(4)}(t)} = \frac{\sum_{z_1, \ldots, z_4} |\Phi_{\text{con}}(z_1, \ldots z_4, t)|}{\sum_{z_1, \ldots, z_4} |\Phi(z_1, \ldots z_4, t)|}. \tag{5}$$

This quantity vanishes for a Gaussian state and meaningfully quantifies its non-Gaussian character. Note that given that $M^{(4)}$ as defined above is a non-negative quantity, in an actual experiment it can only go as low as a small non-vanishing value, reflecting the experimental statistical fluctuations (finite size of the statistical sample). The summation window is typically taken over a region around the middle of the system where the atomic gas is to a good approximation homogeneous and the data is more reliable than closer to the edges. In all scans of the present study the trap used to confine the atoms is box-like, more specifically, a superposition of a harmonic trap with a box trap of much smaller extent so that the density is approximately homogeneous over the entire system. Therefore, unlike for the more commonly used harmonic traps, there is no significant inhomogeneity in our experimental system, except for boundary effects that are still present in the close vicinity of the edges of the box. As a technical remark, in Fig. 3 for scans 1-4 the box-like trap is 75 μm long and we analyze using Eq. (5) the central region of length 50 μm and for scans 5-7 we analyze the central 38 μm region of a 50 μm long trap.

The experiment has observed the decay of this measure for all initial conditions (and trap geometry), as shown in Fig. 3 presenting the dynamics of $M^{(4)}$ for various initial states.

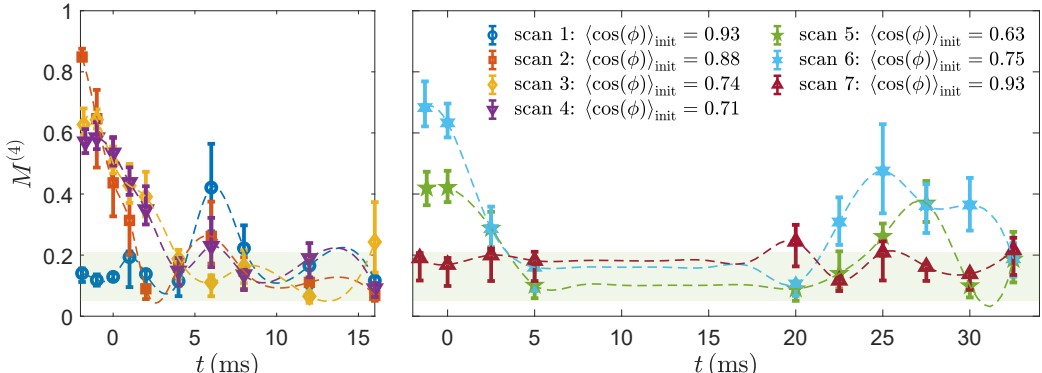

Figure 3: Observation of Gaussification (*left*) and the recurrence of non-Gaussianity (*right*) in the experiment. Plots of the non-Gaussianity measure $M^{(4)}$ as a function of time in various experimental scans corresponding to different initial states, as presented in Ref. [24] (the plots are based on the data published in Ref. [36]). The time window ranges from the initial time until 16 ms *(left)* or 32.5 ms *(right)*. Notice that the decay of non-Gaussianity follows a relatively sharp decrease to a small saturation value. At the recurrence time [30] the non-Gaussianity returns to quite a high value relative to the initial one. The green shaded area indicates the bias values of $M^{(4)}$ corresponding to Gaussian states with finite statistics.

The quantity $M^{(4)}$ decreases rapidly to a small value that is indistinguishable from that of a Gaussian state. Note that, because $M^{(4)}$ is by definition a non-negative quantity and because of the statistical fluctuations in a finite sample of measurements, the mean value of $M^{(4)}$ corresponding to measurements on a Gaussian state is not zero but positive: this is called 'finite statistics bias'. Interestingly, for a box-like trapping geometry, at a particular later time, $M^{(4)}$ becomes large again. This revival of non-Gaussianity has been discussed above and can be fully accounted for by the mechanism of Gaussification by canonical transmutation which will be introduced in more detail in the following section.

A characterization of the initial states based on the strength of the interactions is presented in Fig. 4 where the observed value of $M^{(4)}$ in the initial equilibrium state is plotted against the observed coherence factor $\langle \cos \phi \rangle$. The latter is controlled by the interaction coupling $J$ and unlike that it is directly measurable in the experiment, so that it can play the role of the control parameter for the strength of interactions. For $J$ increasing from zero to a relatively large value the equilibrium coherence factor changes from zero to one. It should be mentioned that in the experiment $J$ cannot be chosen to be larger than $\hbar \omega_{\mathrm{tr}}$ where $\omega_{\mathrm{tr}}$ is the transverse trap frequency, otherwise transverse mode excitations are present. In this case the system can no longer be considered as effectively one-dimensional. The initial state of scan 1 corresponds to the heavy mass regime of the sine-Gordon model (KG regime) where $\langle \cos \phi \rangle \approx 1$ and is therefore Gaussian. Quenching to the TLL regime results in a highly squeezed initial state due to the fast change of the effective mass parameter from a large value to zero. Scans 2, 3 and 4, on the other hand, correspond to non-Gaussian states at decreasing values of the interaction strength $J$. The initial non-Gaussianity reaches its highest observed value in scan 2 and progressively decreases in scans 3 and 4. A theoretical explanation of the observed $M^{(4)}$ versus $\langle \cos \phi \rangle$ curve is given by the classical field approximation of the sine-Gordon model [22] using a stochastic method developed in Ref. [27] whose results are consistent with numerical simulations of the actual quantum model [37].

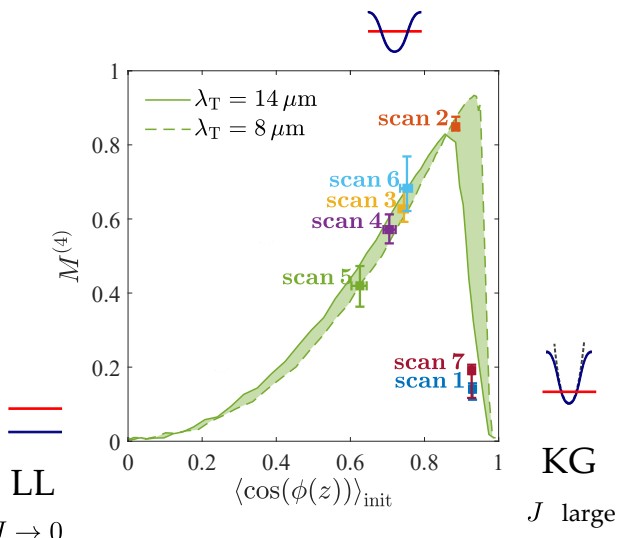

Figure 4: Characterization of the initial states of the experimental scans of Ref. [24]. Plot of the observed value of $M^{(4)}$ in the initial equilibrium state as a function of the observed coherence factor $\langle \cos \phi \rangle$ as first explored in Ref. [22]. The initial state of scan 1 and 7 corresponds to large $J$, i.e., the Klein-Gordon (KG) regime and is Gaussian, while those of scans 2 to 6 are non-Gaussian, with the highest value of non-Gaussianity reached for scan 2. For small $J$ the system is in the Tomonaga-Luttinger liquid regime (TLL). The green curves correspond to theoretical simulations of the sine-Gordon model at thermal equilibrium ($\lambda_T$ represents the thermal coherence length) based on the classical field approximation [27].

## 2 Theoretical discussion of two readings of Gaussification

In what follows, we will consistently refer to *Gaussification* when we speak about the memory loss effect wherein non-Gaussianity decays in an isolated system following an interaction quench in time. It is a main theoretical insight of this work that there are basically two readings of this effect, which we will study and complement with each other in detail in this section. The actual experimental findings from cold atomic quantum field systems will later be put into the context of these Gaussification mechanisms, and we will develop and elaborate on what the actually dominant effect in the experiment is. In what follows, we do not aim at providing a mathematically rigorous framework of Gaussification as presented in Refs. [5, 7, 14, 19], but instead keep the discussion on an intuitive, physically minded level.

### 2.1 Gaussification by spatial scrambling

The first mechanism, and this is what this section will focus on, can be seen as being Gaussification by spatial scrambling. The basic idea to understand is that: Correlation functions of fields delocalized over regions much larger than the correlation length are effectively Gaussian. Expanding on this statement is the goal of this subsection. We keep the discussion close to the experimental setting at hand. In this context, let us consider the field in the above statement to be the particle velocity field, defined as (see Appendix A)

$$\hat{u}(z) = \partial_z \hat{\varphi}(z).$$  (6)

This is a local field, hence one would expect that its correlations cluster in typical initial conditions. This is the first requirement of Gaussification by spatial scrambling constituting one of the pillars in Fig. 1.

**Condition 1** (**Clustering of correlations**). *We assume the initial correlation functions to be exponentially decaying in the distance as*

$$\left| \left\langle \prod_{i=1}^{n} \hat{u}(z_i) \right\rangle_{con} \right| \leq const \times e^{-\frac{1}{2} \sum_{i,j=1}^{n} |z_i - z_j|/\xi} \quad (for \ z_i \neq z_j). \tag{7}$$

*States that satisfy this bound will be referred to as exhibiting exponentially clustering correlations.*

This nomenclature is motivated by the fact that connected correlation functions are substantial only for $z_i$'s that cluster together within a range of the order of the correlation length $\xi > 0$. This property is ubiquitously valid in many systems, specifically in ones prepared in ground states of gapped quantum systems and for those close to thermal equilibrium at sufficiently high temperatures. Whenever all the positions $z_i$ are sufficiently far from each other

$$\left\langle \prod_{i=1}^{n} \hat{u}(z_i) \right\rangle \approx \left\langle \prod_{i=1}^{n} \hat{u}(z_i) \right\rangle_{\text{Wick}}. \tag{8}$$

For general lattice models with finite-dimensional constituents, it has indeed been proven that not only ground states [38], but also high temperature states exhibit a clustering of correlations [39, 40]. Thus, initial states exhibiting exponentially clustering correlations can be strongly correlated states within the range of the correlation length, while exhibiting approximately Gaussian correlations at long distances. The correlations for points separated beyond the correlation length may intuitively be thought of as playing the role of a Gaussian 'bath' in the sense that Gaussification occurs as the result of the dynamical mixing of the initially non-Gaussian component of correlations into the much larger Gaussian component.

The delocalization of the field can result in Gaussianity both at long and short distances as compared to the correlation length. Let us see how this works by means of a simple example. Consider the field integrated over a region of the system $R$

$$\overline{\Delta \hat{\varphi}} = \frac{1}{\sqrt{|R|}} \int_R dz \ \hat{u}(z). \tag{9}$$

Here we weigh this expression by the size $|R|$ of the region. This is similar to considering independent identically distributed variables $X_i$ and forming the central limit variable $\overline{X} = n^{-1/2} \sum_{i=1}^{n} X_i$. Classically, the inverse-square-root normalization of $\overline{X}$ is crucial and then the distribution becomes Gaussian in the limit $n \to \infty$. We can estimate the connected part of this observable as

$$\left\langle \overline{\Delta \hat{\varphi}}^4 \right\rangle_{\text{con}} \approx \frac{1}{|R|^2} \int_{R^4} d^4 z \left\langle \prod_{i=1}^{4} \hat{u}(z_i) \right\rangle_{\text{con}} \approx const \times \frac{\xi^3}{|R|}, \tag{10}$$

where the constant is bounded by the maximum of the correlations at short distances determined by the UV cut-off. We hence see that if we take a field whose correlations exponentially cluster, then the average over a large region results in connected functions scaling inversely proportional to the size of that region. Thus in the limit of $|R| \to \infty$ the connected part vanishes.

A type of such delocalization can be implemented by unitary Gaussian dynamics of an isolated quantum system [5, 7, 16–20]. This is the second pillar of spatial scrambling in Fig. 1.

**Condition 2** (**Delocalizing dynamics**). *The dynamics generated by the non-interacting Hamiltonian is expected to be delocalizing. Said in different words, the propagator of an initially local field decays with time in the entire space.*

Let us expand on this condition in more detail. For a Hamiltonian that is non-interacting, i.e., quadratic in terms of a set of fields, the Heisenberg equations of motion that determine the unitary time evolution of these fields are linear differential equations with respect to time. As such they can be solved for general initial conditions and the time evolved fields are expressed as linear combinations of the initial fields. Conversely, if the unitary time evolution is of this linear form then the governing Hamiltonian must be non-interacting. In the present context, for example, the Heisenberg equation of motion satisfied by $\hat{\varphi}(x,t) = \hat{U}^{\dagger}(t)\hat{\varphi}(x,0)\hat{U}(t)$ with $\hat{U}(t) = e^{-iHt}$ is a linear partial differential equation of second order with respect to time, or equivalently a linear system of two first order equations for the relative phase field $\hat{\varphi}(x,t)$ and its canonically conjugate field $\delta\hat{\varrho}(x,t)$ which expresses the relative density fluctuations of the two-component atomic gas (see Appendix B for details). The solution to the associated initial value problem can be expressed in terms of $\hat{\varphi}(y,0)$ and $\partial_t\hat{\varphi}(y,0) \propto \delta\hat{\varrho}(y,0)$ by means of the Green's function method, which finally gives the field $\hat{u}(x,t)$ expressed as

$$\hat{u}(x,t) = \int dy \; G_u(x,y,t)\hat{u}(y,0) + \int dy \; G_\rho(x,y,t)\delta\hat{\varrho}(y,0). \tag{11}$$

Here $G_u(x,y,t)$ and $G_\rho(x,y,t)$ are the retarded Green's functions, commonly called propagators, corresponding to the equations of motion for $\hat{u}(x,t)$ under the initial conditions $\hat{u}(y,0) = \delta(x-y), \delta\hat{\varrho}(y,0) = 0$ and $\hat{u}(y,0) = 0, \delta\hat{\varrho}(y,0) = \delta(x-y)$, respectively. They encode the dynamics triggered by the initial conditions and express the impulse response of the field $\hat{u}(x,t)$ coming from a localized disturbance of $\hat{u}(y,0)$ or $\delta\hat{\varrho}(y,0)$, respectively. In practice, they can be calculated by expanding in the eigen-modes of the Hamiltonian (Fourier modes in the translationally invariant case).

If the dynamics admits a bound of the form

$$|G_{u/\rho}(x,y,t)| \leq \text{const} \times t^{-\alpha}, \tag{12}$$

for some $\alpha > 0$, then we say that it features delocalization, as summarized in Fig. 1. For example, in a typical lattice system with local dynamics the propagators are upper bounded as above with an exponent $\alpha = 1/2$. The causality of the time evolution (Lieb-Robinson bound) is implemented in $G$ as exponential decay with the distance outside the light-cone region $|x-y| < ct$, where $c$ is the speed of sound. The light-cone propagation bound is also a characteristic of the low-energy effective theory describing the present continuous system. These properties imply then that $\hat{u}(x,t)$ equals to an effective average (with an oscillatory weight function) of $\hat{u}(y,0)$ within the light-cone. This is similar to the above discussed case with $|R| = ct$ and so we expect a similar bound on the connected part of the correlations

$$|\langle \hat{u}(x,t)^4 \rangle_{\text{con}}| \approx \text{const} \times \frac{1}{ct}. \tag{13}$$

We hence encounter the following scenario: If a system exhibits dispersive dynamics, then an initially local field spreads in space under the time evolution and its connected correlation functions should then decay at least as a power-law in time. If instead the dispersion relation is linear, then information about initial correlations travels coherently and non-Gaussianity may be preserved.

It should be remarked that the dynamics of lattice models does not implement precisely a square-root decay of the Green's function in the entire space [5, 7, 19, 20]. Rather, as can be seen, e.g., by the stationary phase approximation, there is a wave-front at the edge of the effective causal cone which decays more slowly than is true for the interior of the cone. This complication is one of the reasons why a complete derivation of Gaussification by spatial scrambling goes beyond the intuitive picture sketched here.

More specifically, in the above we have considered for illustration purposes a typical case of dynamics characterized by an effective light-cone spreading. In non-relativistic continuous systems, however, correlations can in principle spread quite fast with no effective maximum velocity bound. Even in this case where field spreading is not constrained the Gaussification by spatial scrambling arguments may still be applicable [17]. In the illustrative discussion of the mechanism we assumed the scaling exponent to be everywhere in space equal to 1/2 which implies uniform decrease of the Green's function inside the light-cone. This is instructive for a general discussion but again, the delocalization mechanism is valid more generally. In particular, it holds for dynamics generated by local Hamiltonians where the Green's function scales according to the 1/2 exponent in the bulk of the effective light-cone but at the wave-fronts at its edge the asymptotic scaling in time is 1/3 [17,19,20]. In any case the essential arguments available in the literature in all cases can be argued to rely closely on the delocalization of the fields as explained here (see Ref. [20] for an accessible discussion of these complications along the lines that we presented here and Ref. [19] for a general derivation of such types of scaling for local lattice models).

## 2.2 Gaussification by canonical transmutation

Gaussian correlations can also emerge based on a mechanism which does not involve spatial scrambling. Instead of that, we want to consider the possibility of the decay of non-Gaussian correlations that results from the internal dynamics in each of the independent harmonic modes.

To be specific, let us consider the dynamics generated by a general non-interacting Hamiltonian

$$\hat{H} = \tfrac{1}{2} \sum_{k=1}^{\infty} \hbar(\delta\hat{\rho}_k^2 + \omega_k^2 \hat{\phi}_k^2), \tag{14}$$

with canonical commutation relations $[\delta\hat{\rho}_k, \hat{\phi}_{k'}] = i\delta_{k,k'}$ for $k, k' = 1, 2, \dots$ and $\delta_{k,k'}$ is the Kronecker symbol. The time evolution of each harmonic mode under this non-interacting Hamiltonian is

$$\hat{\phi}_k(t) = \cos(\omega_k t)\hat{\phi}_k(0) - \frac{\sin(\omega_k t)}{\omega_k}\delta\hat{\rho}_k(0). \tag{15}$$

Here, we find that the density sector is being rotated into the phase sector and vice versa, which is what we mean by canonical transmutation: In particular, for $t = \pi/2\omega_k$ we have that $\hat{\phi}_k(t) \propto \delta\hat{\rho}_k(0)$ with a complete transmutation of the role of the operators in the canonically conjugate pair. This is the crucial qualitative insight and captures the essential physical process occurring after the quench, which allows us to account qualitatively for the resulting Gaussification dynamics and at the same time is largely independent of the model as even perturbed Hamiltonians would feature similar rotation dynamics. The discussion of this quadrature rotation makes specific the general idea of the necessary dynamical pillar as listed in Fig. 1.

**Condition 1** (**Quadrature rotation**). *The dynamics generated by the non-interacting Hamiltonian is expected to be implemented by a quadrature rotation to generate transmutation.*

This relation is enough to argue that marginals of non-Gaussian states with a certain structure of correlations are going to become Gaussian over time. To give the basic idea, consider the case that initially only the phase sector is non-Gaussian, e.g.,

$$\langle \hat{\phi}_k(0)^4 \rangle_{\text{con}} > 0. \tag{16}$$

At time $t = \pi/2\omega_k$ we have

$$\langle \hat{\phi}_k(t)^4 \rangle_{\text{con}} = \langle \delta\hat{\rho}_k(0)^4 \rangle_{\text{con}} = 0 . \tag{17}$$

We hence see that the connected correlation function of the phase mode $k$ will become fully Gaussian under this transmutation. Here, we see that if there is such a structure of Gaussian correlations being contained in one sector of the canonically conjugate pairs, then we can identify them again as constituting a Gaussian bath. As suggested by this illustrative case, we wish to promote this property (one canonical sector being Gaussian and the other non-Gaussian) to a feature that can also be present in other systems with possibly different degrees of freedom. For this reason we single out this characteristic of the initial state as a distinct pillar of the mechanism of Gaussification by canonical transmutation as listed in Fig. 1.

**Condition 2** (**Existence of a Gaussian canonical sector**). *The initial state should have a canonical sector that is Gaussian and decoupled from the non-Gaussian sector.*

This basic observation so far phrased for essentially the non-local eigen-modes of the system is at play in more complicated situations. First, what happens at intermediate times? Assuming the Gaussianity of one canonical sector, we see that the non-Gaussianity of the other sector of each eigen-mode will oscillate between the initial value and zero. Secondly, such transmutation dynamics within the eigen-modes will lead to more complicated dynamics in real-space. For this let us take the instructive example of the mode decomposition of the fields in a homogeneous system of length $L$

$$\hat{\varphi}(z) \sim \sum_k \cos(\pi z k/L)\hat{\phi}_k . \tag{18}$$

Using Eq. (15), we see that the dynamics of the real-space field $\hat{\varphi}(z,t)$ will lead to a transmutation into the density sector in each mode at its own frequency. For this reason the transmutation dynamics will turn out to be a mixture of some eigen-modes which happened to have rotated significantly and some which did not, resulting overall in a reduced value of non-Gaussianity for almost all times. The analysis of mode mixing is more generally relevant in the study of signatures of space-time propagation of phase correlations and has been studied in Ref. [41].

**Condition 3** (**Dominance of the Gaussian canonical sector**). *The initially Gaussian canonical sector should be much larger than the non-Gaussian one.*

As we will discuss later, the decay of Gaussianity can be substantial if the Gaussian bath is at high temperature and there is an imbalanced energetic penalty which leads to strong squeezing, i.e., $\langle \delta\hat{\rho}_k^2 \rangle \gg \langle \hat{\phi}_k^2 \rangle$. In this case even if the dynamics is weighting both eigen-mode operators similarly, the Gaussian correlations will dominate, because there will be more contribution from fluctuations of the Gaussian type at any moment. When this condition holds, after the quench one would see that fluctuations overall increase with an increasing portion of these correlations being Gaussian. This feature is an important aspect of the effect in the experiment in Ref. [24] and will be dissected in a number of sections below. However, the discussion geared towards the experiment can be viewed as being a feature that should be recognized in its own right as potentially playing a role also in other systems. For this reason, we include this feature as one of the pillars of Gaussification by canonical transformation and refer to it as 'dominance of the Gaussian canonical sector' (Fig. 1.b). Let us also remark that, even though the dominance requirement means that the ratio of initial fluctuations of the Gaussian over the non-Gaussian sector should be much larger than unit, as we will see below, we actually obtain a large decrease of non-Gaussianity even for relatively moderate ratios.

## 2.3  Similarities and differences of the two mechanisms

The two mechanisms discussed in this work have in common the setting in which they can come into play and the type of effect they cause. In both cases, we are interested in a non-Gaussian state influenced by some interactions in the initial Hamiltonian. After interactions have been quenched, the connected correlations become less prominent over time. So both mechanisms can lead to the emergence of Gaussian correlations. One of the most intriguing similarities is that effectively in both cases 'Gaussianity' is not being dynamically produced, but rather dynamically 'uncovered'.

In the case motivated by the experiment, the sector of phase observables has been non-Gaussian while the canonically conjugate operators have been Gaussian. Then, any Hamiltonian that mixes the two sectors sufficiently strongly (quadrature rotation pillar) can lead to Gaussification. Here, however, the extent of Gaussification is given by the extent in which the density correlations dominate over time the phase sector. If this is very substantial then connected correlations will become less important and yield a negligible relative contribution to the full correlation function. From Eq. (15) it seems that the ultimate extent of Gaussification is given by the average between the two sectors which is the result one obtains by a time average over a period $T \sim \omega_1^{-1}$.

One can think of an analogy with two water tanks, one with a high and the other with a low level of water so that after connecting them the levels oscillate and even out. Here, the phase sector has a large degree of non-Gaussianity while the density sector has a low level and both start mixing after the quench such that the full correlation functions of phases become mostly Gaussian due to the Gaussian density fluctuations that are dynamically rotated in after the interaction quench. So here the density correlations play the role of a Gaussian 'reservoir'.

One can identify a type of such reservoir that is dynamically mixed in also in the Gaussification by spatial scrambling. In this case, however, a field becomes spread over a large portion of the system and as we saw earlier higher order correlation functions become largely made of initial correlation functions from positions separated by large distances. These are initially Gaussian as the correlation function factorizes due to the assumption of clustering of correlations. So far away correlation functions can be identified to play the role of the Gaussian 'reservoir' and the spreading of correlations in form of homogeneous spreading of wave-packets 'couples' the local correlations to that reservoir such that local expectation values are described by a Gaussian state to an increasingly better approximation.

Having said that, there are also crucial differences. In Gaussification by canonical transmutation, the non-Gaussianity need not necessarily disappear fully, but rather become only overwhelmed by rotating in other Gaussian correlations. Here, the degree of Gaussification by canonical transmutation is limited by how much densities dominate typically over time, and how Gaussian they are in the first place. Via wave-packet spreading, if the system is asymptotically large, the local observables can be captured arbitrarily well by a Gaussian state, so in contrast to Gaussification by canonical transmutation there is no limit to the convergence to Gaussianity.

Gaussification by canonical transmutation can occur when spatial scrambling will fail to be present, as exemplified by the experiment in Ref. [24]. The most important example where this happens is in systems characterized by a linear dispersion relation, which will not exhibit spreading of wave-packets, so the only possible type of Gaussification is the relative one. As described below (Subsec. 3.2), the required condition that an entire sector of initial correlations is decoupled and Gaussian is possible to hold in a broad class of systems prepared in a high temperature state which is described by the classical field approximation. Therefore, the effect is not limited to the sine-Gordon model as e.g. the $\phi^4$ theory should also exhibit similar equilibrium correlations. In the case of linear dispersion systems discussed here, the different nature of the canonical transmutation compared to the spatial scrambling mechanism

translates into a different scaling of the approach to equilibrium in a large system, as explained in Subsec. 5.2. Another feature of the Gaussification by canonical transmutation as opposed to spatial scrambling is that the overall amount of non-Gaussianity of connected correlations should be preserved. Lastly, Gaussification by canonical transmutation can occur without the condition of initial clustering which is necessary for the Gaussification by spatial scrambling – as long as there is a Gaussian 'reservoir' sector that is rotated in.

# 3 Characteristics of the initial correlations in the experiment

One overarching observation concerning the two Gaussification mechanisms discussed in this work is the presence of Gaussian correlations in the initial state in some form such that they have initially been inaccessible (since we can only measure directly phase and not density correlations) but then come to view via the dynamics. We will now focus on the different possibilities for some type of correlations to be initially Gaussian. We will begin by analyzing one of the two pillars (as summarized in Fig. 1.a) of Gaussification by spatial scrambling, namely the question of clustering of correlations. The next subsection will discuss a microscopic argument that pertains to the experiment.

## 3.1 Clustering and scaling in initial states

In this section, we will be discussing various characteristics that pertain to the scaling of correlations in the experimental system at thermal equilibrium. The relative phase defined in Eq. (1), whose statistics is the direct experimental observable, is not a local field in a strict sense. This is because it is only differences between the phases at two different points (one of which is the reference point) that have a direct physical interpretation. As a result, the correlations of the field that corresponds to the phase difference between two points do not decrease with the distance but instead increase or saturate, due to the cumulative effect of fluctuations in the intermediate spatial interval. This is in contrast to the typical behavior of the correlations of local fields which decrease with the distance when we consider equilibrium states of local systems. The long-range character of phase correlations has been discussed already in earlier works, see, e.g., Refs. [22, 41–44]. Here we explain this behavior in an alternative way (by deriving the scaling of correlations of the phase field from that of its spatial derivative) and argue that in the non-Gaussian states considered here the growth of phase correlations with the distance is related to the presence of topological excitations in the system.

The space and time derivatives $\hat{u} = \partial_z \hat{\varphi}$ and $\partial_t \hat{\varphi} \propto \delta \hat{\varrho}$ of the quantum phase are expected to be local fields. Connected correlations of such local fields in local quantum states are expected to decay with the distance: In ground states of gapped Hamiltonians the decay of local field correlations is exponential with a correlation length that is controlled by the mass of the lightest quasi-particle excitation [45, 46]. In particular, this behavior holds for the sine-Gordon ground and thermal states in the gapped phase [47].

Let us begin the discussion by noting that the phase is related to fluctuations of the atomic gas. In its microscopic description, the atoms are described by quantum fields with creation and annihilation operators denoted as $\Psi^\dagger(z), \Psi(z)$, and the correlations of such atomic fields should decay in space reflecting the locality of the interactions between the atoms and the fact that finite temperature prevents long-range correlations that would signify order in the system. The fluctuations of the phase field are related to the correlations between the atoms such that the former increase when the latter decrease. For example, in the quadratic harmonic fluid

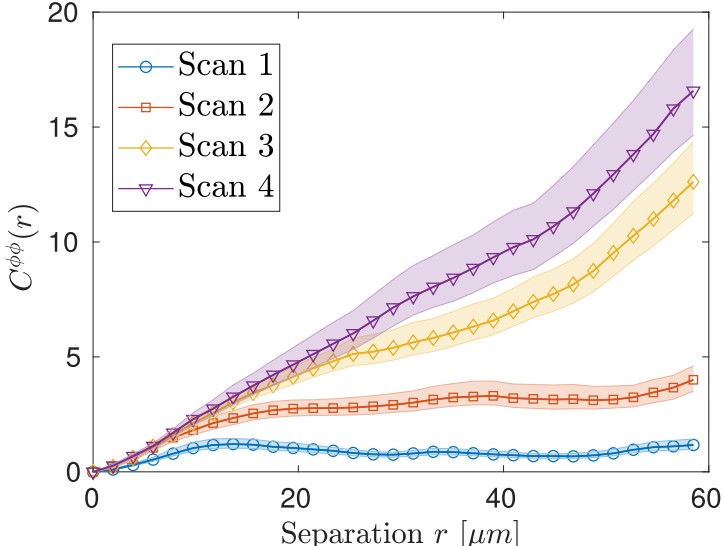

Figure 5: Scaling of phase auto-correlations $C^{\phi\phi}(r)$ as a function of the distance $r$ from the reference point (middle) in the initial state. Scan 1 corresponds to a massive Gaussian state, scans 2, 3 and 4 correspond to non-Gaussian states at different interaction strength. The bands indicate the size of the estimated errors.

approximation

$$g_1(z, z') = \langle \Psi^\dagger(z)\Psi(z')\rangle \propto e^{-\frac{1}{2}\langle(\hat{\varphi}(z)-\hat{\varphi}(z'))^2\rangle}, \tag{19}$$

i.e., the phase difference field is related to the logarithm of the one-particle density matrix of the atoms in the gas [47,48]. The decay of off-diagonal fluctuations of the atoms for increasing distance corresponds to increasing correlations of the phase difference. In the gapless phase of the gas, at finite temperature the relation of fluctuations to correlations is implemented by exponential decrease of $g_1$, i.e., linear increase of phase correlations, while in the ground state the scaling would be inverse algebraic for $g_1$, i.e., logarithmic for phase correlations [48].

In the gapped phase corresponding to the initial states of the experiment, even though the quadratic approximation is not always valid, the relation between the two correlation functions is qualitatively similar. Fig. 5 presents the auto-correlations

$$C^{\phi\phi}(r) = \langle(\hat{\varphi}(z_0 + r) - \hat{\varphi}(z_0))^2\rangle \tag{20}$$

measured in the experiment for initial state preparations with differing barrier heights and hence different strength of the tunnel coupling $J$. We find that over short distances the fluctuations increase seemingly quadratically, eventually switching to a slower increase, corresponding to linear growth or saturation. The rate of the increase depends on the strength of the tunnel coupling.

The cross-over from quadratic to approximately linear scaling can be interpreted by relating to atom fields as alluded to above. On this level we find that the short-range correlation decay of $g_1$ is rapid and governed by a Gaussian function while at large separations we find exponential decay of correlations. The former may be non-universal, affected by the effective cut-off of the field theory implemented by finite measurement resolution. Additionally the short-range correlations may depend on *renormalization group (RG)* irrelevant terms in the Hamiltonian. On the other hand the long-range scaling are expected to be robust and depend primarily on RG relevant terms. This is what we find as the tunnel coupling is effectively described by the sine-Gordon interaction (discussed in detail below) which is RG relevant.

In order to further interpret the scaling of the initial correlations, we can think of the $C^{\phi\phi}$ function based on the local velocity field correlations

$$C^{\phi\phi}(r) = \int_{z_0}^{z_0+r} \mathrm{d}x_1 \int_{z_0}^{z_0+r} \mathrm{d}x_2 \langle \hat{u}(x_1)\hat{u}(x_2)\rangle \,. \tag{21}$$

This integral representation can be further simplified by the following physical considerations. Given that the state is characterized by the presence of massive excitations of the sine-Gordon model in combination with the fact that the velocity field is local, its correlations are expected to decay exponentially with some correlation length $\xi > 0$. Under this assumption, as $r$ becomes larger than $\xi$ we reach a linear scaling of $C^{\phi\phi}(r)$. To see this, it is instructive to switch to the coordinates $x_\pm = x_1 \pm x_2$. The integral of the off-diagonal direction $x_-$ should give a constant roughly proportional to the correlation length. The remaining integral over the diagonal direction along $x_+$ should yield a scaling proportional to $r$ because we are integrating a constant function. Together, this results in a linear scaling

$$C^{\phi\phi}(r) \approx \text{const.} \times r \,, \tag{22}$$

for $r \gg \xi$. Inspecting Fig. 5 we indeed find that for $r > 10\,\mu\mathrm{m}$ a linear behavior is valid, namely for all prepared initial conditions we see either a non-trivial linear increase (scans 3 and 4 corresponding to relatively small $J$) or a leveling-off (scan 1 corresponding to the largest $J$ and to a lesser extent scan 2 at an intermediate value of $J$) compatible with a small or essentially negligible slope constant. In the next paragraph, we will further elaborate on this argument and explain how one can understand the saturation of the correlations $C^{\phi\phi}(.)$ for large coupling using the phenomenology derived from the Gaussian KG field theory.

### 3.1.1 Quantum field simulation of the relativistic field theory models

The state preparation in the experiment involves open system dynamics due to cooling by evaporation or atom losses [49] which yields initial conditions closely matching thermal theory [25, 44], consistent with predictions that can be derived within *Tomonaga-Luttinger liquid (TLL)* and *Klein-Gordon (KG) model* as special limits of the *sine-Gordon (SG) model*. We will now give more details about this description which allows us to capture the system for the limits of a strong and weak coupling of the adjacent one-dimensional gases as depicted. For a high double well barrier, the coupling between the two wells vanishes and the effective sine-Gordon description reduces to the Gaussian TLL model. For a low double well barrier, on the other hand, we are in the limit of large coupling and the description becomes again effectively Gaussian. This can be seen in terms of a semi-classical description: the system lies at the bottom of a very steep cosine potential which can therefore be approximated by a parabola. In this case the effective description is given by the KG theory (see the experimental study [22] and the numerical theoretical analysis [37] for detailed discussions on the crossover from the TLL to the KG regime of the SG model).

Following Refs. [12,22,33], we consider the effective field theory model describing the fast decoupling of two adjacent one-dimensional gases of neutral atoms at ultra-cold temperatures (see Fig. 2 for a schematic of the experimental setup). At strong tunnel coupling between the two wells, the effective model is given by the Hamiltonian

$$\hat{H}_{\mathrm{KG}}(J) = \int \mathrm{d}z \left[ \frac{\hbar^2 n_{\mathrm{GP}}(z)}{4m} (\partial_z \hat{\varphi}(z))^2 + g(z)\delta\hat{\varrho}(z)^2 + J n_{\mathrm{GP}}(z)\hat{\varphi}(z)^2 \right], \tag{23}$$

involving relative fluctuations in phase $\hat{\varphi}(.)$ and density $\delta\hat{\varrho}(.)$ [30, 50]. These low-energy degrees of freedom represent the phononic excitations of a one-dimensional Bose gas and

satisfy bosonic commutation relations $[\delta \hat{\varrho}(z), \hat{\varphi}(z')] = i\delta(z - z')$. Here $m$ is the atomic mass and $n_{GP}(.) \approx$ const is the mean density profile. The phononic operators are defined within the atomic cloud whose spatial extension is given by the support of $n_{GP}$. Lastly, $g(.)$ is the density-broadened interaction strength [30, 51] and the parameter $J$ is the tunnel coupling which is tuned by the double-well barrier height.

The last term of the Hamiltonian that involves $J$ can be viewed as a mass term. Ramping up the barrier height in the experiment, effectively quenches the mass term from a nonzero value of $J$ to zero, so that only the first two terms that make up the TLL Hamiltonian remain, i.e.,

$$\hat{H}_{KG}(J = 0) = \int dz \left[ \frac{\hbar^2 n_{GP}(z)}{4m} (\partial_z \hat{\varphi}(z))^2 + g(z)\delta \hat{\varrho}(z)^2 \right] = \hat{H}_{TLL}. \tag{24}$$

Note that the above effective description in terms of a quantum quench from the massive KG to the massless TLL model is valid only under the condition that the density $n_{GP}$ is constant in both space and time. This holds to a very good approximation in the experimental system when a box-shaped external trap is used, and it is also a sufficiently good approximation in the middle region of the system when a parabolic trap is used instead.

Such a KG to TLL quench is performed in experimental scan 1 and the corresponding measurements of the dynamics will be shown below. Here we will focus on the properties of the initial state preparation. The experiment tends towards the KG regime for very strong tunnel couplings as evidenced here as well. In the KG limit there is strong pinning of the compactified phase field to the minimum of the cosine potential which results in the absence of phase winding, that is, absence of soliton excitations. This means that the phase difference between the edges of the system is very small

$$\int dz \; \partial_z \hat{\varphi}(z, t = 0) = 0. \tag{25}$$

Accordingly, in scan 1 for which the initial state is in the KG regime we observe that the slope in (22) is close to zero and the large distance asymptotics is saturated to a constant (Fig. 5), in contrast to other scans, especially 3 and 4 corresponding to relatively small $J$, which are consistent with an approximately linear increase with the distance.

Summarizing, the correlations of the phase should be thought of as fluctuations as they govern the melting of the ordering of the atomic system and tend to increase with the distance. On the other hand, the correlations of the derivative field $\hat{u}$ in equilibrium states corresponding to the KG limit should be decaying with the distance as a result of pinning of the phase due to heavy phononic excitations. The scaling observed in the experiment indeed reproduces the phenomenology that correlations of the derivative field $\hat{u}$ decay with the distance.

### 3.1.2 Velocity field correlations in the initial state

In this subsection, we are interested in the scaling, in particular, the range of correlations of the local velocity field in the initial equilibrium states, in order to assess the presence of one of the 'pillars' of the spatial scrambling mechanism. Extracting the velocity field two point correlations

$$C^{uu}(z_1, z_2) = \langle \partial \hat{\varphi}(z_1) \partial \hat{\varphi}(z_2) \rangle \tag{26}$$

directly is impossible in the experiment. Instead we look at the approximation obtained by looking at the correlations of phase differences between adjacent pixels

$$C^{uu}(z_1, z_2) \approx \delta z^{(-2)} \langle (\hat{\varphi}(z_1) - \hat{\varphi}(z_1 + \delta z))(\hat{\varphi}(z_2) - \hat{\varphi}(z_2 + \delta z)) \rangle. \tag{27}$$

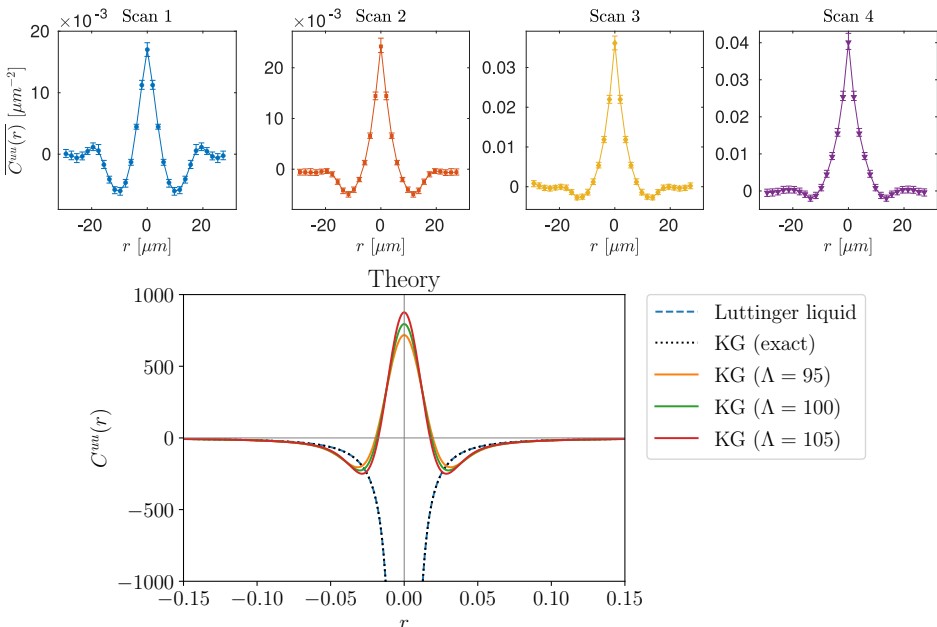

Figure 6: *Top*: Longitudinally averaged velocity field correlations $\overline{C^{uu}}(r)$ for state preparations used in Ref. [24]. From left to right the tunneling strength is reduced. For all scans we see an auto-correlation peak for $r \approx 0$. We see that in the KG regime (scan 1) there is a rapid decay of the auto-correlation switching over to negative values, i.e., an anti-correlation of the velocity field at a distance of about $10\,\mu\text{m}$ which is about 20% of the system length. As $J$ decreases the anti-correlation dip becomes more shallow, eventually vanishing almost entirely. For all scans the velocity field 2-point functions are consistent with exponential fall-off envelopes with a length-scale which is a small portion of the system size. *Bottom*: Theoretical plots of the correlation function of the velocity field $C^{uu}(r)$ as a function of the distance $r$ in the Tomonaga-Luttinger liquid and Klein-Gordon model (large coupling limit), at one choice of parameter values $c = 1$, $M = 1$, $T = 0$ and various values of the high-momentum cutoff $\Lambda$. We observe the characteristic negative correlation dips on the two sides of the central peak.

In practice, we have $\delta z \approx 2\,\mu\text{m}$ which in typical configurations amounts to 2 to 4% of the system length.

Fig. 6 shows the extracted profiles where we average over all pairs of positions $z_1$ and $z_2$ with a fixed distance $r = z_1 - z_2$. For large $J$, i.e., in the KG regime (scan 1), we find once again a profile consistent with a vanishing integral in the distance $r$, as discussed above, implemented by a strong anti-correlation on the two sides of the central peak that cancels the auto-correlation at zero distance $r \approx 0$. The profile matches qualitatively with the theoretical prediction for equilibrium correlations in the KG model, also plotted in Fig. 6 for comparison with the experimental plot for scan 1. The theoretical formula for the distance dependence of the correlations in the KG model in a large homogeneous system is (see for example [52])

$$C^{uu}_{\text{KG}}(r) = -\partial_r^2 C^{\phi\phi}_{\text{KG}}(r), \qquad C^{\phi\phi}_{\text{KG}}(r) \propto \int_{-\infty}^{+\infty} \mathrm{d}k\, \frac{e^{ikr - \frac{1}{2}(k/\Lambda)^2}}{E(k)} \left( \frac{1}{2} + n_{\text{BE}}(E(k), T) \right), \qquad (28)$$

where $E(k) = \sqrt{c^2 k^2 + M^2 c^4}$ and $n_{\text{BE}}(E, T)$ is the Bose-Einstein distribution at temperature $T$. The parameter $M$ is the KG particle mass, which is $M \sim \sqrt{J}$ here, and $\Lambda$ is a high-momentum cutoff. At large distances the above correlation function decays exponentially with

a correlation length determined by the mass $M$ and temperature $T$, while at short distances the theoretical scaling is the same as in the TLL ground state, $M = 0$, $T = 0$, which is $\sim -1/r^2$. However, the contribution of high momentum modes is suppressed by the cutoff $\Lambda$, which in the experiment is dominated by the finite imaging resolution which has the form of a Gaussian weight function [33]. As a result the profile of the correlation function switches from a singular function to the shape shown in Fig. 6. The precise height and width of this profile is controlled by all parameters $M, T$ and $\Lambda$. For this reason and given that a precise estimate of $M$ and $T$ is not available, a quantitative comparison of the experimental and theoretical plots is not possible, however, the qualitative behavior is similar.

Decreasing $J$ to intermediate values in the sine-Gordon regime, the anti-correlation dips are weakened and the integral over $r$ becomes nonzero in agreement with the linear increase of $C^{\phi\phi}(r)$ at large distances observed in the previous section. In the context of the sine-Gordon model the physical meaning of the velocity field, i.e., the phase derivative, is the density of solitons in the system. A non-vanishing value for the correlations of the velocity field integrated over a spatial region signifies fluctuations of the number of solitons in that region. This is precisely what one would expect for equilibrium states of the sine-Gordon model in the strongly correlated regime. Therefore the above scaling analysis provides an indirect signature of the presence of solitons and of their thermodynamics in the experiment, already observed in the full counting statistics of the phase field in Ref. [22].

## 3.2 Canonical decoupling into Gaussian and non-Gaussian sectors

It is often the case that the internal dynamics of a physical system can be very well described by a Hamiltonian consisting of two parts where individually each part involves only a commuting set of operators and the non-commutative, i.e., quantum, character of the model comes from the fact that the two parts are non-commuting. This is very well illustrated by the phononic degrees of freedom that we have in mind where typically, e.g., for equilibrium conditions at low temperatures we have the generic form

$$\hat{H} = \hat{H}_\phi + \hat{H}_\rho \,. \tag{29}$$

Individually, the thermal state of only one part of the canonical pairs, so $\hat{H}_\phi$ or $\hat{H}_\rho$ would agree with a classical probability distribution, but in general for $\hat{H}$ this need not be true. Interestingly, at sufficiently large temperatures the bosonic statistics of the degrees of freedom is expected to become less prominent. Whenever this is true, such an effect makes sure that the thermal state of the entire Hamiltonian

$$\sigma_\beta = e^{-\beta\hat{H}}/Z_\beta \,, \qquad Z_\beta := \mathrm{tr}[e^{-\beta\hat{H}}] \tag{30}$$

agrees closely with a classical probability distribution. To illustrate this point, in the most extreme case of very high temperatures, we have

$$\sigma_\beta \approx (1 - \beta\hat{H})/Z_\beta \,. \tag{31}$$

Here, we see that the correlation function of either the phase or density operator depends on its respective part of the Hamiltonian and so the non-commuting aspect of the fields does not even have a chance to play a role.

This observation coming from the simple high-temperature expansion may be valid only at temperatures much higher than in the experiment. However, to make the extrapolation to lower temperatures plausible, one should notice that as the temperature increases, the energy of the system must also increase, so we must have that $\langle\hat{\phi}_k^2\rangle$ and $\langle\delta\hat{\rho}_k^2\rangle$ must also grow with temperature. This means that the occupation numbers of the modes involved will become

increasingly larger than the vacuum level. However, only for states close to the ground state the canonical commutation relations play a prominent role because the expectation value of the commutator operator, which is independent of the state and proportional to $\hbar$, is comparable to the mode occupation numbers.

This leads to the *classical field approximation (CFA)* being a good approximation, where we assume that the phase and density operators are effectively commuting with each other. This suggests that we can effectively 'trace-out' one of the sectors of the operators without quantum corrections, so by treating the phononic fields as independent classical fields not coupled with each other. For the lack of a better term, we will refer to such an independence of the correlation functions of fields in each canonical sector from the canonically conjugate sector as canonical decoupling. Whenever such a feature is at play we can still make use of further observations to plausibly infer characteristics of the correlations of the quantum many-body system considered.

Specifically, let us assume that one of the canonical parts of the Hamiltonian is a quadratic form of the degrees of freedom. Anticipating the relation to the experiment, let us take the density part to be quadratic while the phase part can include a many-body potential going beyond the quadratic term, as in the sine-Gordon model. Under the assumption that the two canonically conjugate fields are effectively decoupled for the temperature in question, we obtain a very crucial prediction, namely that the density-density fluctuations can be approximated as

$$\langle \delta \hat{\varrho}(z_1) \dots \delta \hat{\varrho}(z_n) \rangle \approx \mathrm{tr}\left[ \delta \hat{\varrho}(z_1) \dots \delta \hat{\varrho}(z_n) e^{-\beta \hat{H}_\rho} \right] / Z_\beta^{(\rho)}, \tag{32}$$

where again $Z_\beta^{(\rho)} = \mathrm{tr}[e^{-\beta \hat{H}_\rho}]$ for $\beta > 0$. This correlation can then be obtained using Wick expansion of the only non-trivial correlation function, i.e., the second moments

$$\mathcal{Q}(z,z) = \langle \delta \hat{\varrho}(z) \delta \hat{\varrho}(z') \rangle. \tag{33}$$

This expression makes specific what we mean by tracing out: The correlations in the density sector are taken to be described by the thermal state of the density Hamiltonian $\hat{H}_\rho$. We take the same temperature as we would take for the full thermal state of the full Hamiltonian. The former observation should be stressed as the independence of tunnel coupling means that at all its values $\hat{H}_\rho$ is Gaussian so actually the correlations in the density sectors should be Gaussian too. This means that the density fluctuations should be *independent* of the tunnel coupling and approximately thermal. In summary, the argument suggests that, crucially, CFA implies that at sufficiently high temperatures *all* higher-order moments in the density sector should be *Gaussian*. Thus the Gaussian bath in the canonical transmutation mechanism is a result of canonical decoupling and at least one of the canonical sectors being Gaussian.

By allowing for tunneling of atoms between the adjacent gases an effective interaction between the phonons becomes relevant which can give rise to kink excitations according to the *sine-Gordon* (SG) model whose Hamiltonian is given by

$$\hat{H}_{\mathrm{sG}} = \hat{H} + J \int_{-L/2}^{L/2} \mathrm{d}z \; n_{\mathrm{GP}}(z) \cos(\hat{\varphi}(z)) . \tag{34}$$

Here $J/(2\pi\hbar)$ is the single particle tunneling rate, which can be tuned by the barrier height.

Having this specific model in mind, lets us state what the implications of canonical decoupling would be in our system. First, the factorization of the thermal state density matrix at high temperature means that the density correlations would be given by expectation values computed in a state $\sim e^{-\beta \hat{H}_\rho}$ which is a diagonal quadratic form of the density field. This

means that all higher-order functions of the density would be Wick contractions based on the two-point function which at high-temperatures reads

$$\mathcal{Q}(z, z') \approx (\beta g)^{-1} \delta(z - z').$$  (35)

The value of phase correlation functions can in turn be computed as explained in Ref. [27] by replacing the quantum phase operators by classical phase variables since the canonically conjugate operators should not play a role at high temperatures, so only commuting operators would be involved. Put differently, we would take the phase correlation functions to be the correlation functions in the classical sine-Gordon model.

In the large temperature limit the correlation length of the density fluctuations is very small. This in particular should imply that to a good approximation we can write for eigen-mode operators

$$\langle \delta \hat{\varrho}_k^2 \rangle = \text{Const.}$$  (36)

We can recover this qualitative feature by starting from the formula for thermal correlations of a set of eigen-modes

$$\langle \delta \rho(x_1) \delta \rho(x_2) \rangle_0 \propto \sum_{n=1}^{\infty} [1 + 2 n_{\text{BE}}(E; \beta)] E(k_n) e^{i k_n (x_1 - x_2)},$$  (37)

where $n_{\text{BE}}(E; \beta) = (e^{\beta E} - 1)^{-1}$ is the Bose-Einstein distribution at inverse temperature $\beta > 0$. In the above the eigen-mode energy is denoted by $E(k_n)$, while $c$ and $K$ are the sound velocity and Luttinger parameter, respectively (see Appendix A for details). We now see that for $\beta^{-1} \to \infty$ we have $n_{\text{BE}}(E; \beta) \approx 1/(\beta E)$ from which we find

$$\langle \delta \rho(x_1) \delta \rho(x_2) \rangle \approx \text{Const.} \times \delta(x_1 - x_2).$$  (38)

By inspecting formula (37), we see that in the Gaussian case we will see quantum corrections, related to vacuum fluctuations, only once the first order expansion of the exponential in the Bose-Einstein distribution becomes an inadequate approximation. This sets a scale for the temperature for Gaussian CFA to be closely linked to the eigen-mode frequency of the low-lying modes observable in the experiment. To account for such short-range correlations we would indeed see that

$$\langle \delta \hat{\varrho}_k^2 \rangle \approx \text{Const.}.$$  (39)

Let us remark that in the case of an interacting system, an approximate Gaussian description at large temperature would include effects of the interaction through the self-energy corrections [52]. This asymptotic behavior is generally valid independently of the precise form of $E(k)$ and therefore independent from the interaction which enters only through the self-energy insertions in $E(k)$. In the present case, the above result justifies (36) and explains why that constant is independent of the sine-Gordon tunnelling $J$.

Summarizing the entire section, here we have argued that phase and density decouple at high temperatures if there are no terms in the Hamiltonian directly involving both types of operators. In particular, this would imply that density fluctuations should be Gaussian in the SG model which seems to be the case for temperatures relevant in the experiment. This is the canonical decoupling pillar summarized in Fig. 1. Of course, this property alone does not imply that the Gaussian density sector dominates the phase sector during the dynamics. The next section explains that this is actually expected to be the case for the experimental system.

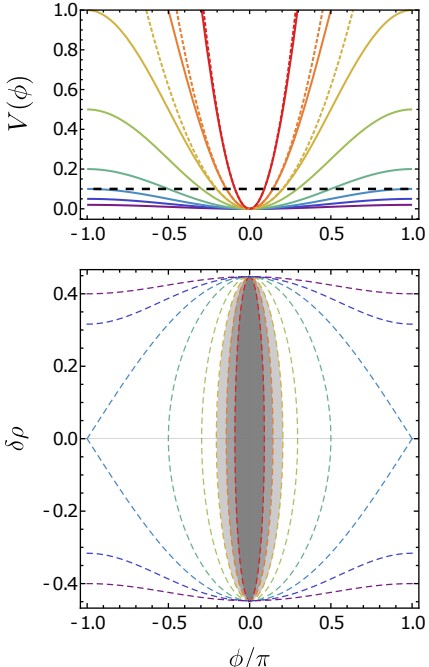

Figure 7: Semi-classical explanation of the dominance of density over phase fluctuations in the sine-Gordon model. Ignoring the spatial variation of the phase, the sine-Gordon model can be approximated as the quantum analogue of the classical pendulum [53]. *Top:* At a fixed temperature (corresponding to mean total energy denoted by the dashed black line), increasing $J$ results in a steeper and narrower potential well that can be approximated by a parabola (dashed colored lines). *Bottom:* Seeing from a phase-space point of view, increasing $J$ results in stronger 'squeezing' of the wave-function, illustrated through the elongation of the dashed line contours that represent the energy levels (or the shape of a phase space distribution of a classical thermal ensemble). The values of $J$ from purple to red are 0.01, 0.025, 0.05, 0.1, 0.25, 0.5, 1., 2.5 and the mean total energy level is $E = 0.1$.

### 3.3 Dominance of density over phase initial correlations in the experimental states

#### 3.3.1 Intuitive understanding of the experimental findings

We are now turning to discussing the third 'pillar' of Gaussification by canonical transmutation, which is the condition where the initial correlations of one of the two canonical sectors dominate over those of the other. The correlations of each sector at intermediate times $t > 0$ that are not integer multiples of a half recurrence period will be a mixture of the initial correlations of both sectors. Therefore, if one of the sectors has larger correlations than the other initially, it will dominate in the time evolved correlations of both sectors.

Let us now explain why this can well be the case in the experiment. As already mentioned, the initial states are effectively equilibrium states of the SG model or, in case of large coupling, of the KG model. For clarity, consider the phase and density fluctuations in thermal states of the two limiting cases of the SG model, i.e., the KG and TLL models, (23) and (24) respectively, which are quadratic. In the TLL model expressed as a sum of decoupled modes (14), we know from the equipartition theorem that the energy contributions of each of the two quadratures to the total energy are equal

$$\langle \delta \hat{\varrho}_k^2 \rangle = \omega_k^2 \langle \hat{\phi}_k^2 \rangle \qquad \text{(TLL equilibrium states)} . \tag{40}$$

Similarly, in the KG model equipartition means that

$$\langle \delta \hat{\varrho}_k^2 \rangle = \omega_{0k}^2 \langle \hat{\phi}_k^2 \rangle \qquad \text{(KG equilibrium states)}, \tag{41}$$

where $\omega_k = ck, \omega_{0k} = \sqrt{c^2 k^2 + M^2 c^4}$ are the mode frequencies of the TLL and KG model respectively with $M$ the effective mass of the KG excitations, which increases with the coupling as $\sqrt{J}$. From this relation, we find that because the coupling $J$ is large, the density fluctuations $\langle \delta \hat{\varrho}_k^2 \rangle$ are much larger than the phase fluctuations $\langle \hat{\phi}_k^2 \rangle$. In physical terms, this is due to the energetic penalty imposed by the steep parabolic KG potential on the phase fluctuations. The same argument applies on equilibrium states of the SG model for any nonzero value of the coupling $J$ since the energy is related to the correlations of the density and phase field and there is an energetic penalty on phase fluctuations only. The underlying semi-classical argument is illustrated in Fig. 7. More generally, for any model in which there is an energetic penalty, induced by a potential, which applies on *only* one of the two canonically conjugate fields (here, the phase), the corresponding fluctuations will be suppressed compared to those of the other.

In the experiment, there is an initial penalty on phase fluctuations due to the tunnel-coupling $J$, while the density fluctuations are free to fluctuate according to the given temperature. Hence, their amount will be larger than the fluctuations in the phase sector. By this argument, we find that the phase correlation function (initially non-Gaussian) should increase in magnitude after the quench in the form of an increase of the Gaussian component.

### 3.3.2 Estimation of density fluctuations via phonon tomography

In the experimental setting considered, only one quantum field quadrature is directly experimentally accessible (phase), but not the canonically conjugate quadrature (density). Using the quantum read-out method of Ref. [25], however, we can estimate the content of density fluctuations as suggested above. In this procedure, we reconstruct the full covariance matrix of the initial unknown state by relating the non-equilibrium second moments of the phase $\Phi(z, z', t)$ at points $z, z'$ and for the time $t \geq 0$ through the known TLL evolution equations, collecting all suitable second moments. The results of this most simple theoretical model agree very well with the experimental observations.

In our case, the phase correlations $\Phi(z, z', t)$ defined in Eq. (2) can be measured at a discrete set of points $z, z'$ (with a spacing given by the pixel size of the camera $\delta z \approx 2\,\mu\text{m}$ [33]) at time various $t \geq 0$. Apart from the pixel size, other effects, including diffraction, limit the spatial resolution. The measured values can be related to theoretical continuum predictions by implementing a real-space cut-off via a Gaussian convolution with standard-deviation $\sigma \approx 3.5\,\mu\text{m}$. From Eq. (15), we see that for a sufficiently large number of pixels and time snapshots, the eigen-mode correlations of the phase can be extracted first and from them the corresponding eigen-mode correlations of the density through fitting the dynamics to those predicted by phase space rotation, making a quantitative reconstruction feasible. The implementation of the tomographic reconstruction [25] extends and optimizes this intuitive idea using *convex optimization techniques* to find the full covariance matrix (including density correlations) $\Gamma$ at the initial time $t = 0$ such that the corresponding forwards propagation $\Gamma(t)$ exhibits phase correlations matching the observed data, under the constraints arising from the Heisenberg uncertainty principle. This approach makes a lot of sense in the light of the fact that the Heisenberg uncertainty principle for covariance matrices sets a lower bound on quadrature fluctuations, giving rise to a semi-definite and hence convex constraint. In Ref. [25], the functioning of this method has been laid out in detail and convincingly demonstrated in a study of effectively Gaussian state dynamics with large initial tunneling and quenching to zero tunneling [30]. Here, we apply it to derive the second moments of the (generally, non-Gaussian) quantum initial states corresponding to the present quench parameters.

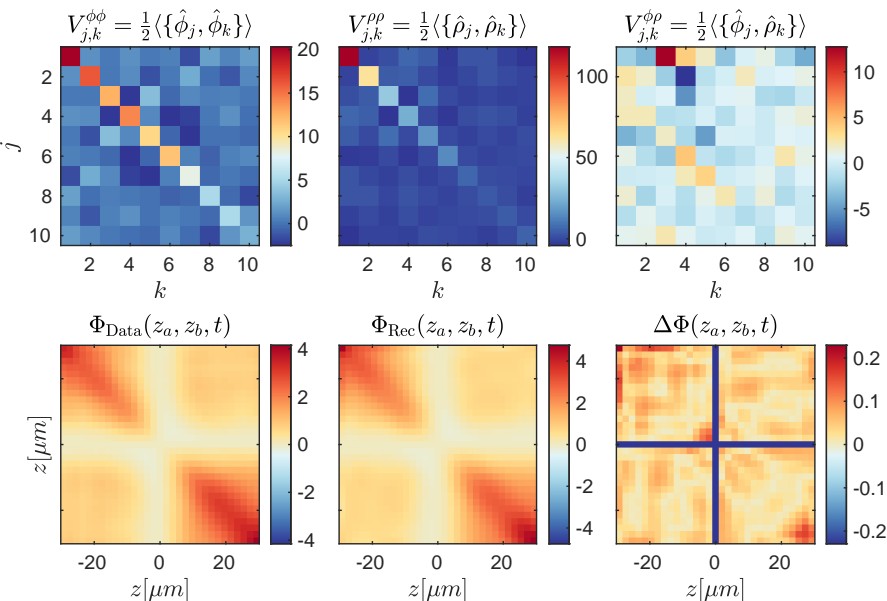

Figure 8: *Top:* Reconstructed initial state correlations in the eigen-mode basis. *Bottom:* The tomographic recovery is performed by matching the measured phase second moments at times $t = 0, 1, \ldots, 12$ ms. Here, we show the comparison for the initial time. The cost function $\Delta\Phi$ is the relative deviation of the reconstructed second moments $\Phi_{\text{Rec}}$ from the data $\Phi_{\text{Data}}$, weighted by the estimate of the root-variance at the given point.

Fig. 8 shows the application of the tomographic recovery of second moments. We find that the eigen-mode quadratures, defined through the rescaled phase and density mode variables

$$\tilde{\hat{\phi}}_k = \hat{\phi}_k \sqrt{\omega_k}, \qquad \delta\tilde{\hat{\rho}}_k = \delta\hat{\rho}_k / \sqrt{\omega_k}, \tag{42}$$

are squeezed in the sense that the diagonal correlations in the density sector $V_{k,k}^{\rho\rho}$ are larger than those in the phase sector $V_{k,k}^{\phi\phi}$. This statement pertains directly only to second moments. The above is further verified at quantitative level in Fig. 9 where the diagonals of the eigen-mode quadrature correlations are plotted for each of the four main scans of the experiment. In all scans, $V_{k,k}^{\rho\rho}$ is larger than $V_{k,k}^{\phi\phi}$ for almost all modes, at least by a factor of 2 in the lowest modes which are the ones that weigh in most in the calculation of spatial averages of correlations. Also note that $V_{k,k}^{\phi\rho}$ is smaller than $V_{k,k}^{\rho\rho}$ and $V_{k,k}^{\phi\phi}$ for almost all modes, as expected for a rapid quench. This is because

$$V_{k,k}^{\phi\rho} \propto -\frac{\text{d}}{\text{d}t}\langle \hat{\phi}_k \hat{\phi}_k \rangle|_{t=0}, \tag{43}$$

which vanishes for instantaneous quenches since there is no dynamics prior to $t = 0$. For quenches of a nonzero ramp duration $\Delta t$ that is short compared to the characteristic time scale of the dynamics, i.e., $\Delta t \ll 1/\omega_k$, the corresponding $V_{k,k}^{\phi\rho}$ is nonzero, given that the time evolution starts already before $t = 0$ at the beginning of the quench ramp, but still small compared to $V_{k,k}^{\phi\phi}, V_{k,k}^{\rho\rho}$. The negative sign of $V_{k,k}^{\phi\rho}$ in most of the scans is due to the fact that the phase correlations increase as a result of the mixing with the density correlations.

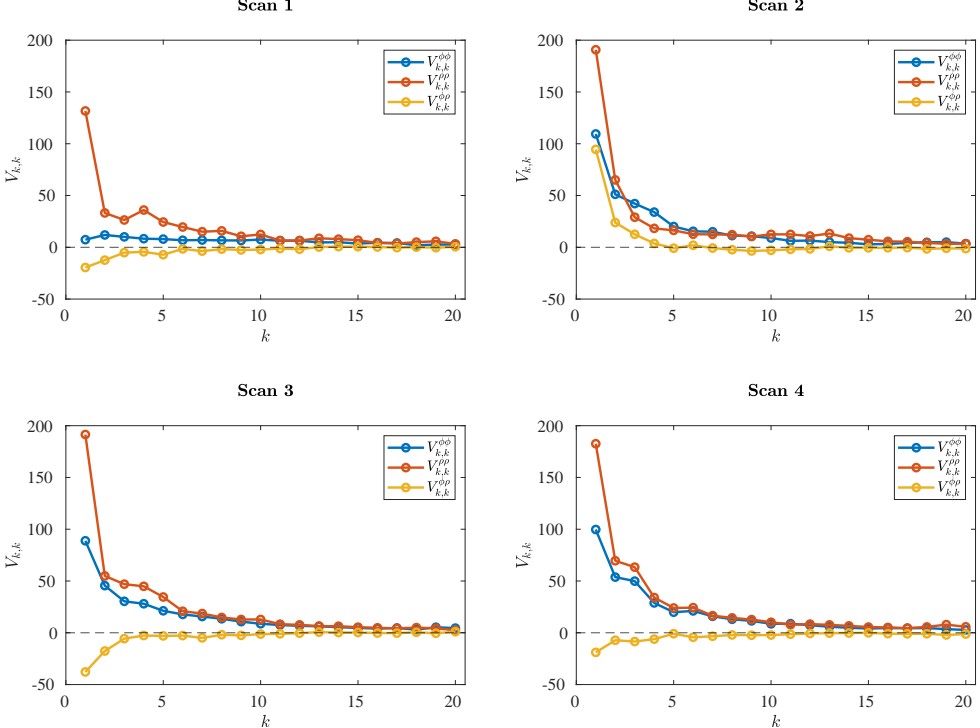

Figure 9: The tomographically reconstructed diagonal quadrature moments confirm the dominance of density over phase initial correlations in the experiment: The plots show the correlations $V_{k,k}^{\phi\phi} = \omega_k\langle\hat{\phi}_k\hat{\phi}_k\rangle$, $V_{k,k}^{\rho\rho} = \omega_k^{-1}\langle\delta\hat{\rho}_k\delta\hat{\rho}_k\rangle$ and $V_{k,k}^{\phi\rho} = \frac{1}{2}\langle\hat{\phi}_k\delta\hat{\rho}_k\rangle$ as functions of the mode number $k$ in the initial state of each of the four scans, as reconstructed by means of the tomographic recovery of second moments of quadratures. We observe that in all scans, $V_{k,k}^{\rho\rho} > V_{k,k}^{\phi\phi}$ at least by a factor of 2 in the lowest modes, and also $V_{k,k}^{\phi\rho} < V_{k,k}^{\phi\phi}, V_{k,k}^{\rho\rho}$ by even an order of magnitude in most cases.

These observations verify quantitatively the accuracy of the fitted model for the dynamics. Moreover, they partially verify the validity of the condition of dominance of density over phase fluctuations in the initial state, as required for the canonical transmutation mechanism. We find that the former are indeed larger than the latter, even though not sufficiently larger to claim that they are dominant. We rather find that their ratio is of order $\sim 2$. Nevertheless, as we will later see in Fig. 15, this magnitude is in agreement with estimates based on the classical field approximation and even though only moderately large it is sufficient to explain the observed decay in the experiment. Importantly, the tomographic reconstruction provides an independent verification that the canonical sectors are sufficiently squeezed as required for the canonical transmutation mechanism, based on directly the experimental data rather than on the classical field approximation.

Let us remark that in principle the tomographic recovery method employed here can be generalized to higher order correlations, but there are two obstructions that make this step largely impractical. Firstly, if the reconstructed correlations are corrupted by noise, they would not necessarily precisely reflect those of a physical quantum state, as is well-known. The constraints that stem from quantum uncertainty bounds on the correlations can only be easily implemented on the level of second moments as a *semi-definite*, that is to say a convex, constraint. This alone does not ensure, however, that the full quantum state is a positive semi-

definite operator. To ask whether such a consistent quantum state can be found is related to the quantum marginal problem, a computational problem that has no efficient solution. This is an obstruction commonly faced when generalizing Gaussian tomography to full quantum state tomography [54]. The second issue is that the available number of experimental runs is limited, in other words, there is a limitation on the maximum size of the statistical sample. As discussed and exemplified in the supplementary material of Ref. [25], even just the second moment reconstruction would profit from larger sample sizes as this reduces the statistical errors on the input second moments and hence improves the precision of the reconstruction. Increasing the number of experimental runs is challenging, however, as it is hard to ensure the stability of the calibration of the experimental setup over extended periods of time. This leads to a significant obstruction for reconstructing higher-order correlation functions: The amount of data to be reconstructed grows exponentially with the order but the sample size and hence precision of the input is constant in practice. This is the reason why we have not attempted the reconstruction of, e.g., the 4-point correlation tensors to assess the question whether the density sector has negligible cumulants.

Fig. 10 displays the reconstructed full covariance matrix to show the second moments of density fluctuations in real space. As shown in Fig. 10, we find that the reconstructed density correlations match closely the density fluctuations obtained from a Gaussian thermal state in the KG model. The thermal theory deviates from the exact computation in continuum (35) due to the application of an appropriate smearing to account for the finite measurement resolution which affects the input to the tomography. More specifically, the tomography is performed over a constant number of the lowest-lying modes, which effectively implements a hard cut-off on the higher energy modes. The eigen-mode wave functions are oscillatory, even when computing them numerically in the presence of inhomogeneities. As is known from quantum field theory – and more generally from harmonic analysis – a Fourier transform of a discontinuous signal has a long-range support. We see this effect also in our case: If we set the occupation of the KG thermal modes to zero above the chosen cut-off, we obtain long-range off-diagonal artifacts. This would not be the case for a smooth cut-off, say an exponential or a Gaussian decay. Nonetheless, we see that the local information, centered around the diagonal, is already converged in shape and magnitude signifying the matching in real space. In the appendix, we present the results for a reconstruction over a doubled-up number of available eigen-modes and we find that the cut-off effect is reduced, with the concentration around the diagonal being increased.

Through this, we have verified the presence of the density fluctuations in the initial state of the system as suggested above. The most significant finding is the verification based on quench data of the presence of a large amount of density fluctuations in the system. This is depicted in Fig. 8 and 9, where we encounter the situation that the values of $V^{\rho\rho}$ are significantly larger than $V^{\phi\phi}$, similar to the results obtained earlier in Ref. [25]. We then go beyond previous works and show that the off-diagonal correlations in the reconstructed eigen-mode covariance matrix do not lead to significant deviations from a diagonal real-space covariance matrix of the diagonal correlations expected from thermal theory. This is shown in Fig. 10, where by comparison to a theoretical computation with a hard cut-off we see a very close matching.

Additional plots on tomographic reconstructions are presented in Appendix D.

# 4 Characteristics of the dynamics in the experiment

While we have so far focused on discussing properties of the initial quantum states, we now turn to elaborating on the actual dynamics in the experiment. The ultra-cold gas in the experiment can be thought of as a fluid which responds to local external disturbances



Figure 10: Comparison of the reconstructed initial second moments of the density with the theoretical predictions for a thermal state (at estimated parameter values $J = 0.5\,\text{Hz}$ and $T = 60\,\text{nK}$). The density plots show the correlations in coordinate space in the entire system (length $L = 60\,\mu\text{m}$). The reconstructed correlations agree quite well with the theoretical ones. The first two plots show an approximation of the continuum limit correlations corresponding to a very fine discretization and computing the respective eigen-modes of the TLL model, while the second two show a discrete grid of points corresponding to the experimentally measured data, i.e., a coarse-grained representation of the correlations with pixel size $\sim 2\,\mu\text{m}$. We find that the transformation of the reconstructed eigen-mode correlations to real space yields a reliable comparison around the diagonal. The off-diagonal features can be directly linked to the presence of a maximum mode cut-off and hence do not represent true correlations in the system.

forming waves. These wave-packets are phonons and their effective models, TLL and SG, have been discussed above. The experiment reproduces the phenomenology of the equilibrium conditions captured by these models making it an excellent candidate for a quantum simulation platform of continuous quantum fields. Out of equilibrium the situation can be markedly different, however, because the universality of thermal equilibrium properties rooted in the renormalization group theory can be distorted by previously irrelevant terms that may affect significantly the dynamics. This section will discuss various qualitative aspects that are important features for developing a complete phenomenological understanding of the non-equilibrium dynamics in the system.

## 4.1 Signatures of light-cone dynamics

*Light-cone dynamics* constitutes a crucial feature of the phenomenology of phononic dynamics. The effective Hamiltonians often possess a relativistic form giving rise to an effective Lorentz symmetry parametrized by the speed of sound. Whenever the dynamics is governed by a gapless Hamiltonian which is to a good approximation translation invariant and quadratic, then the low energy spectrum is linear, i.e., wave-packets propagate practically without dispersion. This is the characteristic property of TLLs and it is responsible for the coherent propagation of particle and hole excitations at low energies in the interacting one-dimensional Bose gas. This dispersion-less propagation of wave-packets is generally sensitive to deviations from homogeneity, phononic interactions and other perturbations. Nevertheless, the effects of such perturbations in the experiment do not become important before a relatively long time passes, which means that the dynamics is practically dispersion-less for the entire duration of the experiment. This is what allows us to observe a clear light-cone wave-front in the propagation of phase correlations.

Light-cone dynamics have been first studied experimentally on the atom chip in Ref. [41] on the level of the phase field. The propagation velocity (*speed of sound*) can also be extracted by studying the appearance of recurrences for different system sizes [30], the very appearance of which has been enabled by making the trap homogeneous [55]. This can be even

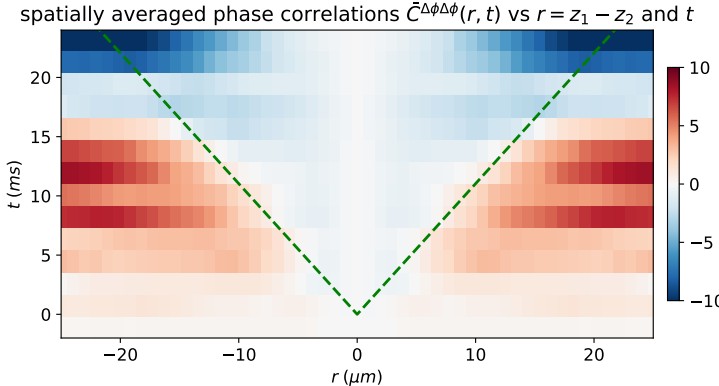

spatially averaged phase correlations $\bar{C}^{\Delta\phi\Delta\phi}(r,t)$ vs $r = z_1 - z_2$ and $t$

Figure 11: Phase difference correlations $\overline{C^{\Delta\phi\Delta\phi}}(z_1, z_2, t)$ averaged over the mean position $z = (z_1 + z_2)/2$ and plotted as a function of the distance $r = |z_1 - z_2|$ and time $t$. The presence of a light-cone shape which separates the outside region where correlations are largely flat as initially from the inside region where they almost instantly switch to their equilibrium values is clearly visible (see also Ref. [41] for a detailed discussion). The dashed green lines correspond to the theoretical estimate for the sound velocity as captured by Eq. (58). The averaging, which is done over all pairs of points at the same distance $r$ in the middle region of the system, i.e., along lines parallel to the diagonal of the phase covariance matrix, serves to reduce the statistical fluctuations. At a given time, the middle region agrees with the dynamics in a thermodynamically large system. After time $t \sim 15\,\mathrm{ms}$, phase correlations in the middle region of the system start being causally connected to the edges of the system and the dynamics is affected by finite size and boundary effects.

further improved by making use of the innovation in the atom control brought about by the implementation of the *digital-mirror device* [56, 57] that allows for programmable potentials in space and time.

A density plot demonstrating the light-cone propagation of phase correlations in the experimental system is shown in Fig. 11, which presents the dynamics that ensues following an interaction quench from a very strongly tunnel-coupled state of two adjacent gases to an effectively decoupled system of two independently evolving systems. As discussed above, the initial Hamiltonian and the one after the quench involve two extreme regimes of tunnel-coupling values and for this reason both systems are effectively Gaussian [22]. The parameters of this experimental scan are similar to those of scan 1, with the difference that the box trap size is larger ($100\,\mu\mathrm{m}$) and the time snapshots are equally spaced at steps of $2\,\mathrm{ms}$. The observation of the shape of the correlations wave-fronts and estimation of their width as a function of time is rather challenging due to the non-trivial profile of the initial correlations of both the phase and density fluctuations, which are mixed together and parallelly transported through the system. Here, the fact that the initial correlations of the local derivative fields are of short range especially in the limit of large initial coupling $J$ allows us to observe the shape of the light-cone edges which follow quite straight lines as expected for a homogeneous system. Even though it is hard to tell if the wave-fronts spread with time or not, which is what is crucial for the spatial mixing mechanism, we can verify that their spreading is negligible at least for times up to half a recurrence period. This is exactly the time window that is relevant for the equilibration stage of the dynamics.

The data presented in the plots carry two principal sources of distortion. The first is due to statistical uncertainties arising from finite sample statistics. Secondly, the measured phase profiles are affected by limitations in the spatial resolution of the experimental read-out [33].

The comparison to a theoretical thermal state becomes possible after applying a Gaussian convolution with variance $\sigma = 3.5\,\mu m$ (this should be compared with the system size which is $L \approx 70\,\mu m$). The observation of light-cone propagation is an indication that the linear phonon dispersion, an otherwise idealized property, is actually an accurate approximation of the dynamics of the experimental system.

## 4.2 Recurrences of the initial state

Another characteristic of the dispersion-less nature of the TLL dynamics is the emergence of recurrences. This is because recurrences fade easily when energy levels deviate from being commensurate. Recurrences in an isolated quantum many-body system have first been observed in Ref. [30] on the level of the coherence phase correlations. Moreover, in Ref. [24] (and as shown in Fig. 3 based on the results of that experiment) it has been demonstrated that the same recurrences can be observed even on the level of higher order correlations, as a more stringent test of returning back to the initial state.

Let us try to better understand why recurrences are linked to the linear dispersion of the dynamical model. First, recall that the energy levels of a TLL confined in a box trap with hard wall boundary conditions (of either Neumann or Dirichlet type) are integer multiplies of a fundamental frequency $\hbar\omega_0 = \hbar c\pi/L$ (see Appendix A), therefore the initial state is expected to be fully recovered after time equal to $2L/c$. Physically, this is the time needed for the lowest-energy phononic excitation to travel around the entire length of the system twice and return back to its initial position in the same direction of motion. At the same time all higher energy excitations travel an integer number of times around the system. At half of this time the lowest excitations have travelled once around the system and the state is a mirror reflection of the initial one, which also results in a recurrence of phase coherence, since given the reflection symmetry of the initial state phase correlations are identical to the initial ones. Therefore the first recurrence is expected at time $T_{rec} = L/c$. In Ref. [30], not only one but two recurrences have been clearly observed, yet exhibiting a decreasing amplitude indicating the eventual loss of coherence in the experiment after a sufficiently long time.

The emergence of recurrences is a property very sensitive to the commensurability of the energy spectrum. More than that, in a TLL the energy levels are equally spaced, a characteristic consequence of the linear dispersion, which makes the observation of recurrences easier. If the above requirement is not met, approximate recurrences are expected, however, the fidelity between the recurrence state and the initial one would be strongly suppressed in the experimental settings. For this reason, it is not by accident that the observation of recurrences was only achieved when it became possible to construct external trapping potentials of a hard-wall box shape [30]. Satisfying this condition required a new technique for the implementation of a blue detuned optical dipole potential. Until then the trapping potential used in the experiment was of the standard parabolic shape, resulting in an incommensurate energy spectrum $E_n = \hbar c\sqrt{n(n+1)}/R$ where $R$ is the Thomas–Fermi radius of the trapped gas. Even though approximate recurrences of coherence are expected for this type of spectrum [58], these are strongly suppressed and so never observed in the experiment. From the above, it should be now clear how fragile recurrences are as an aspect of the dynamics and, for that reason, how accurate witness of the linearity of the phononic dispersion relation their experimental observation is.

## 4.3 Deviations from the non-interacting phonon dynamics in the experiment

The effective field theory model describing the dynamics in the system on the level of non-interacting models constitutes a good approximation, but is nonetheless an approximation after all. Here, we will discuss it in general, naming the possible deviations that may be

relevant in the experiment. The model at hand can actually be derived by a low energy approximation of the *Lieb-Liniger model* by looking at the fluctuations around the classical density and phase profiles obtained from the *Gross-Pitaevskii equation*. Hence, one obtains a hierarchy of effective models each containing terms including only up to a fixed number of phase and density fluctuation operators. The first non-trivial order is quadratic and yields an effective description of the system in terms of non-interacting phonons. As we have shown, this first order approximation matches quantitatively the experimental observations up to intermediate time scales, but for long evolution times various deviations start becoming visible in the experiment. One aspect of the dynamics that is generally sensitive to deviations is the recurrences that occur because the system is isolated and the phonons have a sufficiently linear spectrum [30]. The occurrence of recurrences quantitatively matches with the predictions of the quadratic phononic model, in particular in terms of the timing (recurrence period) which is dictated by the speed of sound and length of the system. The former can be controlled by changing the mean atom density and the latter by choosing the width of the box-like potential. For the experimentally accessible values the quadratic model accounts for the observed recurrence times. Furthermore, recurrences are not visible for non-homogeneous systems which arise, e.g., from harmonic trapping, again in accordance with this model [55].

While the overall timing of the recurrences seems to be robust, the quality of the recurrences deteriorates over time [30]. This damping may be accounted for by fluctuations in the atom number in each experimental realization, which result in fluctuations of the speed of sound, by deviations from homogeneity of the experimental mean density profile, or by higher-order interaction effects present in a stochastic Gross-Pitaevskii model [30]. On the experimental level, a tomographic analysis has shown that the damping of the revivals is closely connected with a steady and irreversible reduction of the thermal squeezing in the eigen-mode quadratures. During this squeezing damping process there are only small indications of the overall growth of phonon occupation numbers. This suggests that the higher-order terms responsible for this effect are irrelevant under the renormalization group as the equilibrium features remain intact, while out-of-equilibrium ones are not. A small drift can also arise from a linear coupling between the relative and the symmetric phase-density sectors of the two coupled condensates, which, however, is expected to be less substantial than the non-linearity effects.

There are a number of additional effects that seem to play a negligible role and are discussed in detail in Appendix C. These include in particular the possibility of wave-packet spreading due to phononic dispersion non-linearities. One source of such effects which is significant at large wavelengths occurs when the trap is not homogeneous. For harmonic traps, in particular, the energy dispersion is $\omega_n \propto \sqrt{n(n+1)}$, i.e., the spectrum is non-linear for the lowest eigen-modes, which correspond to large wave-lengths. As the spectrum converges for large $n$ towards a linear function, like in the homogeneous trap case, sufficiently compact wave-packets are unaffected by this non-linearity and do not disperse substantially enough to play a key role in the Gaussification dynamics. Hence, we conclude that even in the case of dynamics in a parabolic trap the effect is not strong enough in the experiment to facilitate the delocalizing dynamics pillar of the Gaussification by spatial scrambling mechanism.

The second possibility for a non-linearity that could potentially lead to dispersion of wave-packets comes from the finite healing length of the atomic gas. The effect of this on the spectrum can be accounted for by the Bogoliubov dispersion relation, which gradually deviates from linear at small momenta to parabolic at high momenta. This deviation enters through the Galilean kinetic energy part of the Bose gas energy. However, as discussed in Appendix C, the $k^3$ correction to the linear dispersion relation $\omega_k = ck$ is negligible compared to the momentum cut-off dictated by the read-out spatial resolution and its spreading effect is also negligible for the duration of the experiment. Again, while cubic perturbations to the spectrum

could potentially induce a dynamical delocalization effect as needed for the spatial scrambling mechanism, we find that this does not play any role in the experiment.

Finally, the effective boundary conditions at the edge of the system can affect the dynamics. While Neumann conditions seem to be the physically most soundly motivated, since they correspond to a vanishing particle current at the edges, the fact that the density profile falls off smoothly rather than sharply means that the assignment of an effective length for the experimental system cannot be unambiguously done at arbitrary precision. For this reason, it is reasonable that the right effective boundary condition of the system is of mixed type with a small admixture of the Dirichlet type. However, for all practical purposes Neumann boundary conditions seem to allow accounting for all observations made.

# 5  Characteristics of Gaussification in the experiment

Before we analyze the above experimental observations into the context of the earlier presented Gaussification mechanisms, we discuss some further characteristics of Gaussification in the experiment and different ways of how one can observe and study it.

## 5.1  Gaussification in the dynamics of full counting statistics

The picture of Gaussification put forth here complements the previous discussion, in that Gaussification is now being considered from the perspective of full counting statistics and its dynamics. Each interferometric measurement in the experiment yields a phase profile which consists of a few dozen data points for typical system sizes. Treating the measured phase profiles as samples of a random variable (classical, because the quantum phase operators at equal times are mutually commuting for any two positions) we can study the time evolution of its distribution by plotting the corresponding histograms at various times.

Fig. 12 shows experimental results for the phase difference distribution function, initially and during the dynamics. More specifically, the histograms correspond to phase differences $\Delta\varphi$ between all pairs of points in the middle part of the system at the same distance $r = 18\delta z$ where $\delta z = 1.952\,\mu m$ is the pixel size, at the same time $t$. We observe that the initial distributions are visibly non-Gaussian in scans 3 and 4: There is a central peak around $\varphi \approx 0$, but also long tails extending far from the peak whose presence renders the distribution non-Gaussian. This is also clear in the log-scale histograms of Fig. 12 which display significant deviations from the inverse parabola of a Gaussian distribution.

The central peak can be intuitively understood by the presence of a strong energetic penalty on phase fluctuations. Because the sine-Gordon potential is of the form $\propto \cos(\hat\varphi(z))$, the weight of the probability distribution of the phase values is expected to be concentrated around integer multiples of $2\pi$, reflecting the presence of soliton configurations. The presence of smooth jumps by $2\pi$ in some of the observed phase profiles is a clear demonstration of soliton configurations in the physical system which are a typical characteristic of the sine-Gordon model. These manifest themselves as satellite peaks in the phase distribution located at $\Delta\phi \approx \pm 2\pi$. Such peaks have indeed been observed when fast cooling is used for the preparation of the quantum states [22]. In the present case, however, where the initial states are prepared by slow cooling, to a good approximation they correspond to thermal equilibrium states of the sine-Gordon model in which soliton configurations turn out to be strongly suppressed for the temperatures of the experiment. Nevertheless, the non-parabolic shape of the cosine interaction is reflected in the presence of the non-Gaussian long tails in the phase histograms of the initial states.

Over time, the deviations from the Gaussian shape relax: As we see in Fig. 12, the probability distributions remain centered around $\varphi \approx 0$ (which is a gauge choice implemented

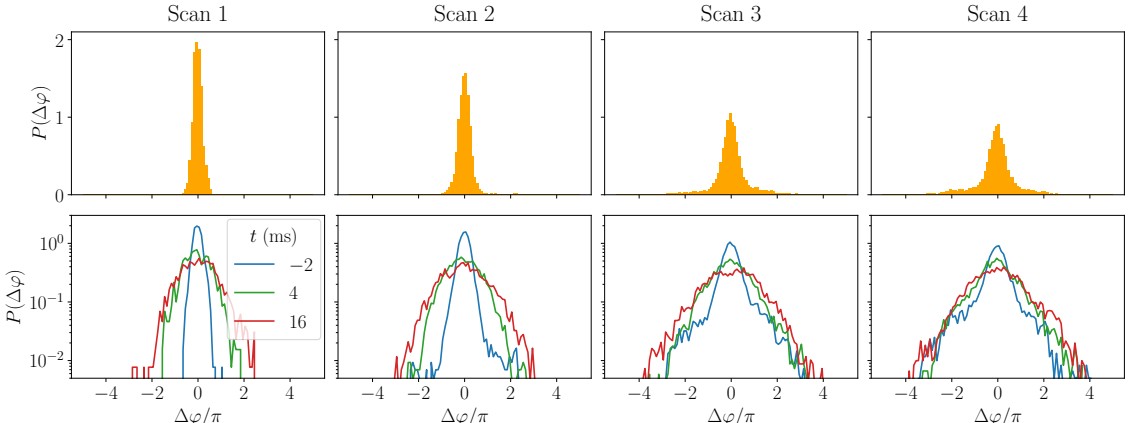

Figure 12: Histograms of the phase difference distributions and their dynamics for each of the four main experimental scans. *Top*: Histograms of the state just before the quench ramp ($t = -2$ ms). Scan 1 corresponds to a narrow bell-shaped distribution that is close to the Gaussian shape. Scans 2, 3 and 4, while still characterized by bell-shaped distributions in the center, progressively deviate from the Gaussian for larger $\varphi$, exhibiting long tails, as one would expect from thermal states of an interacting model with a cosine potential. *Bottom*: Histograms at different times in logarithmic scale: just before the quench ramp, at an intermediate time and at the last measured time. The plots are presented in logarithmic scale to facilitate the identification of the Gaussian bell shape (inverse parabola). The time $t = 0$ corresponds to the end of the quench ramp.

by referencing the profiles) but becomes substantially wider, gradually converging towards a profile closely resembling a Gaussian one. This is quantitatively demonstrated by the time evolution of the moments (the variance and kurtosis) of the phase distribution, as shown in Fig. 13.

The dynamics of the phase distributions can be explained in the same way as the dynamics of the $M^{(4)}$ measure. In the TLL model, which effectively describes the dynamics after the interaction quench, the non-Gaussian features of the initial phase distributions are diluted in the dominant Gaussian bath of the density fluctuations. Additionally, after the quench the phase locking due to the cosine potential is removed resulting in the broadening of the phase

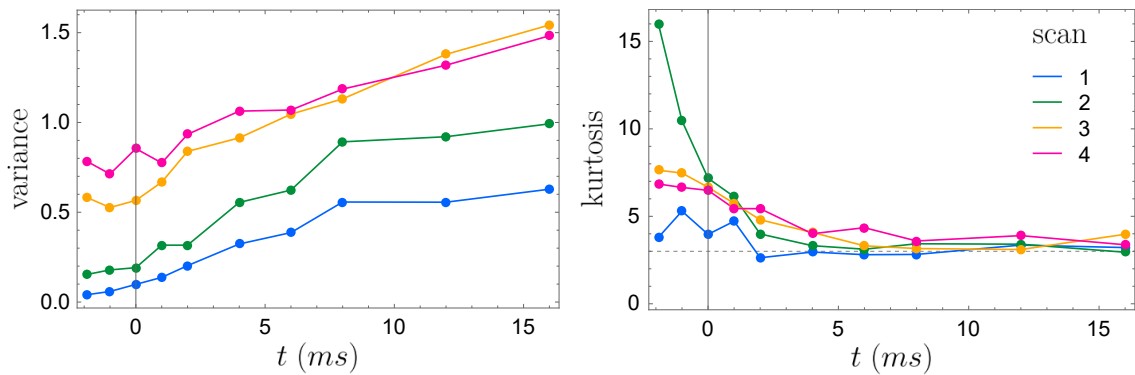

Figure 13: Time evolution of the variance (*left*) and kurtosis (*right*) of the phase difference distribution as a function of time $t$, in the four different scans. The variance increases, while the kurtosis tends to the Gaussian value (i.e., to 3).

distribution. In more detail, because of the phase space rotation the initial phase field is mixed with the initial density field and since the latter shows purely Gaussian fluctuations, is independent from the former and exhibits a much larger variance, the resulting distribution broadens with time. This can be seen from (15) expressed in coordinate space

$$\hat{\varphi}(x,t) = \frac{1}{2}(\hat{\varphi}(x+ct) + \hat{\varphi}(x-ct)) + \frac{\pi}{2K} \int_{x-ct}^{x+ct} \delta\hat{\varrho}(x')\,\mathrm{d}x'. \tag{44}$$

As we see, the fluctuations of the phase field at any time and any position can be traced back to initial fluctuations of the phase and density fields. In a large system, these initial fluctuations are independent from each other because, on the one hand, the phase fluctuation contributions originate from distant points, on the other hand, the density fluctuations are independent from the phase ones. In this case, the distribution of $\hat{\varphi}(.,.)$ is the convolution of those of the independent constituent initial fields. As a result, given the above characteristics of $\delta\hat{\varrho}$, the field $\hat{\varphi}(.,.)$ approaches a Gaussian distribution and broadens with time. The same arguments apply for the statistics of phase differences $\Delta\varphi$.

## 5.2 Temporal properties of the decaying connected correlations

We can understand the temporal onset of Gaussification in the experiment by assuming decaying initial correlations of the velocity field in combination with the CFA properties of the Gaussianity of the density fluctuations and independence from the phase fluctuations. In Appendix B, we show that under these assumptions the magnitude of an $n$-point correlation function will be reduced by factor $2^{-n+1}$. In the case of the 4-point correlation functions which are in focus here, this amounts to almost an order of magnitude reduction of the extent of connected functions in the phase sector.

This reduction of the connected functions is derived from a similar starting point as the considerations leading up to Gaussification by spatial scrambling but using only the first pillar, i.e., correlation clustering for the density and velocity fields, and without assuming validity of the second pillar, i.e., dynamical delocalization. The presence of this second pillar which pertains to dynamics wherein wave-packets delocalize over time and would essentially imply Gaussification by spatial scrambling is not verified in the experiment. In contrast, we will now assume non-dispersive TLL dynamics and hence no delocalization of the correlation wave-fronts which remain concentrated at the edges of the effective light-cone. Intuitively, this consideration shows how the absence of the delocalization pillar affects the physics that would be seen if it was present.

In the appendix, we make the following intuition more precise. While in TLL dynamics there is no delocalization, any local operator will split into two parts which stay local but propagate in opposite directions to the left and right of the initial position. If we then consider a 4-point function of the velocity fields

$$C^u(z, t=0) = \langle \hat{u}(z,0)^4 \rangle_{\mathrm{con}}, \tag{45}$$

then after time $t$ we will find that only $1/8$ of the possible correlation functions resulting from this simple splitting have significant correlations. These are those that precisely overlap in time

$$C^u(z,t) = \frac{1}{8}\langle \hat{u}(z \pm ct, 0)^4 \rangle_{\mathrm{con}} + O(e^{-ct/\xi}), \tag{46}$$

where the second term comprises by all the other contributing terms which are tightly upper bounded in time and become suppressed as long as the size of the light-cone becomes larger than the correlation length $ct \gg \xi$.

Figure 14: Schematic explanation of the equilibration time scaling in the canonical transmutation mechanism. Due to the linear dispersion of the TLL liquid, the phase field propagates according to the wave equation, meaning that it depends on initial data in the interior of the past light-cone (a). However, the phase field is not a physical observable itself: It is only phase differences with respect to some reference point $\Delta\phi(x) = \phi(x) - \phi(x_0)$ (b) or derivatives of the phase field (c) (corresponding to the hydrodynamic fields like the velocity $\hat{u} \sim \partial_z\phi$ and density field $\delta\hat{\varrho} \sim \partial_t\phi$) that are well-defined observables. The propagation of these fields as derived from that of the phase field is constrained to remain localized in the vicinity of the light-cone edges. The time evolved correlations between two (or more) such fields relax shortly after their initial-time spatial supports stop overlapping, which happens when the light-cone edges distance from each other, thus resulting in a linear scaling with time for phase differences and exponential scaling for derivative fields (c.f. Appendix B).

We see that the above arguments result in a prediction for the dynamics of connected correlation functions of the velocity fields. Namely, we expect a linear in time decrease to a plateau value that is reduced compared to the initial correlations. By integrating the velocity correlations this suppression pre-factor will be translated to a decrease of the connected part of phase correlations. This explanation is schematically illustrated in Fig. 14 and in more detail in Appendix B.

The temporal profile of linear scaling followed by saturation is a general feature of the dynamics of correlations in the TLL model. An intuitive explanation is given by the quasi-particle description of quantum quenches. In the particular case of TLL dynamics this interpretation says that the quench creates quasi-particle excitations that travel with the speed of sound $c$ and spread correlations from their initial positions throughout the system [6,59]. If the initial correlations are of short range then this mechanism gives rise to the above described profile and the change from linear scaling to saturation is sharp, otherwise the temporal profile gets smeared. If the dispersion relation is not linear as in the TLL, then the quasi-particles travel with a range of different velocities, which also results in a smearing of the temporal profile. Moreover, it results in a slower equilibration process typically characterized by power-law tails, since the slowest quasi-particles arrive later than the fastest and the transient dynamics lasts longer, consistently with the discussion in Sec. 2.1. These arguments apply also in the case of non-Gaussian initial states and higher-order connected correlation functions [16]. Even though it is harder to estimate the scaling of $M^{(4)}$ given that it is a spatially integrated measure, the same qualitative behavior is valid also here.

We hence obtain an interesting answer to the question what happens if the delocalization pillar of Gaussification by spatial scrambling is missing: We find that non-Gaussianity can still reduce over time, but it does not tend towards zero and is rather reduced by a certain factor. It should be pointed out that this decrease can be significant. Nevertheless, as shown in the Supplementary Information of Ref. [24] based on numerical simulations in the classical fields approximation, this decrease is still insufficient to explain the experimentally observed decrease. Indeed the experimental decrease stops at a much smaller value that was only possible to explain through the canonical transmutation mechanism.

In Fig. 3, we see that the decay of non-Gaussianity corresponds to a rapid decrease to a

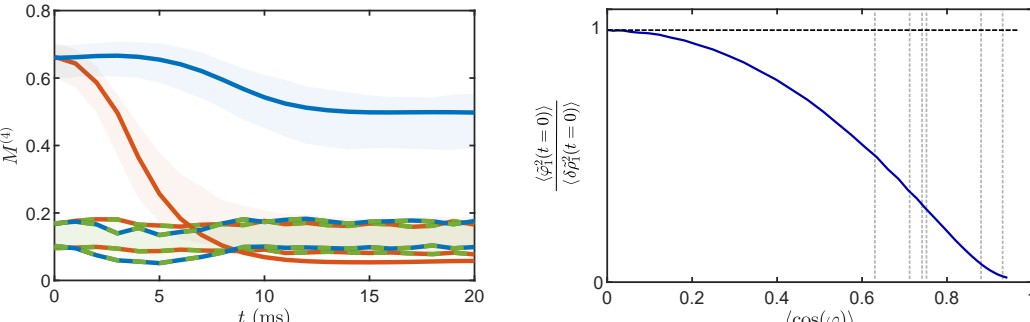

Figure 15: *Left:* Time evolution of $M^{(4)}$ in a theoretical simulation of the experiment for typical parameters. The simulations are based on a stochastic method for the construction of thermal states of the classical sine-Gordon model [27], followed by TLL dynamics. The parameter values of the theoretical model used here have been estimated from the experimental data as discussed in detail in [24]. The red line corresponds to one of the non-Gaussian initial states and its dynamics agrees very well with those observed in the experiment. The blue line corresponds to the artificial dynamics of the same initial state if the contribution of density correlations is ignored. Equivalently, it corresponds to an initial state with the same phase fluctuations but with the initial density fluctuations set to zero. The dashed lines correspond to Gaussian fluctuations with finite statistics and set the bias level. *Right:* CFA estimate of the ratio of phase over density initial fluctuations of the lowest cosine mode, calculated in thermal states of the sine-Gordon model as a function of the coherence factor $\langle \cos \phi \rangle$, which is monotonically increasing for increasing coupling $J$. The gray vertical lines indicate the values corresponding to the experimental initial states (c.f. Fig. 3). We observe that in this parameter range the estimated ratio varies between $1/2$ and close to zero, values consistent with the tomographic estimates of Fig. 9. For higher modes the corresponding ratio is always larger than that of the first mode. This suggests that even for moderately small values of the ratio, the corresponding time evolution shown on the left exhibits a significant decrease of non-Gaussianity comparable to the experimental observations.

level that is practically indistinguishable from mere Gaussian bias due to finite sample size. We find that the profile of the decay of the non-Gaussianity measure $M^{(4)}$ appears to match with the description of a fast linear decrease to a value much lower than the initial one. The decay profile can be contrasted with the power-law decay of the upper bound envelopes obtained via spatial scrambling. Therefore we may conclude that the decay profile is fully consistent with that of the canonical transmutation mechanism rather than the slower decay of the spatial scrambling mechanism. This is further corroborated by theoretical simulations based on the classical fields approximation as shown in Fig. 15. Using parameter values consistent with the experimental data to model the initial states and subsequently applying TLL dynamics on them, we find that the time evolution of non-Gaussianity, in particular the rapid and almost linear decrease to the bias value, agrees very well with the experimental observations. Moreover, if we ignore the contribution of the initial density fluctuations field to the dynamics, we clearly see that, although the non-Gaussianity decreases from the initial value also in this case, it does not reach the bias value but stops decreasing far above it.

What we should stress is that the non-Gaussianity measure $M^{(4)} = S^{(4)}_{\mathrm{con}}(t)/S^{(4)}_{\mathrm{full}}(t)$ shown in Fig. 3 is a relative measure of non-Gaussianity. The absolute value of the integrated four-point connected correlation function $S^{(4)}_{\mathrm{con}}(t)$ in the experiment decreases but does not decay to

almost zero. It is partly this decrease and partly the increase in $S_{\text{full}}^{(4)}(t)$ that results in the drastic decrease of $M^{(4)}$. This is indeed observed and has been pointed out in Ref. [24]. Eq. (46) indicates a linear in time decrease of the non-Gaussianity of connected correlation functions in absolute, not relative size. Note that extracting estimates of higher-order correlations of the derivative fields from experimental data seems futile due to the stronger effects from noise and error sources. For this reason, it has not been possible to use non-Gaussianity measures based on derivative fields in the experiment.

The analysis of Appendix B can explain these qualitative characteristics of the temporal profile of the non-Gaussianity measure $M^{(4)}$ as observed in the experiment and predicted by the numerical simulation of Fig. 15. In particular, it explains the reduced but nonzero value of $M^{(4)}$ at large times when the density correlations of the initial state are artificially set to zero. It also explains why it is only the relative measure $M^{(4)}$ that decays and not the absolute value $S_{\text{con}}^{(4)}(t)$ of the integrated four-point connected correlation function. According to the above, the latter is expected to decrease to a relatively small but considerable fraction of the initial value. The lesson drawn is that, while TLL dynamics do lead to a decrease of non-Gaussianity, which is due to the partial mixing of initial left and right moving field correlations induced by the dynamics, that alone is not sufficient to explain the experimentally observed decrease, which instead can be easily explained by the mixing of phase and density fluctuations according to the canonical transmutation mechanism.

## 5.3 Interpretation of the experimental data

We now turn to a discussion of the experimental findings of Gaussification and how they relate to the two mechanisms laid out above. Let us first summarize our findings regarding the occurrence of the conditions of the two mechanisms in the experiment and then reach what could be called a verdict on which of the two is mainly responsible for the observed Gaussification. While this analysis will not come to a fully consistent conclusion that settles the question in all detail beyond any doubt, the comprehensive analysis put forth here as well as the substantial data taken will paint a pretty clear picture on the mechanisms that can be held responsible for Gaussification in the experiment.

We again would like to start by evaluating the applicability of the spatial scrambling mechanism for the data at hand. We have argued that the initial state exhibits clustering (referred to as mechanism 1, pillar 1, in the above description) only for a class of the physical fields, specifically those that are space or time derivatives of the phase field $\hat{\varphi}$, which correspond to the fluid velocity $\hat{u}$ and density displacement $\delta\hat{\varrho}$. These are the fundamental (local) fields of the hydrodynamic description. A demonstration of the decay of $\hat{u}$ initial correlations with the distance has been shown for the two point function (Fig. 6). The same can be inferred for the $\hat{\varphi}$ initial correlations based on the tomographic reconstruction, as shown in Fig. 10. The phase field $\hat{\varphi}$ itself can only be measured through the difference between two points and therefore its correlations increase with the distance from the reference point. It is no surprise that this field does not exhibit clustering. Moreover, from the observation of relatively sharp light-cone fronts and – more importantly – of strong recurrences of the initial state, we have to conclude that the experimental dynamics do not exhibit delocalization (referred to as mechanism 1, pillar 2 in the above description). This is consistent with the theoretical dynamics of a TLL, which is characterized by a linear phononic dispersion relation. In the absence of this second pillar, the first mechanism results in only a mild decrease of non-Gaussianity that can be attributed to the mixing of left and right moving components of the hydrodynamic fields $\hat{u}$ and $\delta\hat{\varrho}$, as shown in Fig. 15 and discussed in Subsection 5.2.

Having said that, we now turn to elaborating on the canonical transmutation mechanism.

The requirement of quadrature rotation (so mechanism 2 and pillar 1) is automatically satisfied on the basis of the above verification that the system is subjected to TLL dynamics. Based on the classical fields approximation, we have argued that the initial state, modelled as a thermal state of the sine-Gordon model, is expected to exhibit two crucial properties with respect to the correlations of the two canonical fields.

Firstly, the two canonical sectors should decouple, i.e., the correlations of the density and the phase should be independent, which at the same time means that only the latter are non-Gaussian (mechanism 2, pillar 2). Secondly, the density correlations should be larger than those of the phase (mechanism 2, pillar 3) for any value of the coupling $J$ and temperature $\beta^{-1}$. Both of these statements have been verified in the experiment, at least on the level of second moments, using the tomographic reconstruction of the initial state: Density-density mode correlations are indeed generally larger than phase-phase ones, and both are typically much larger than the density-phase correlations (Fig. 8 and 9). Even though the ratio of density over phase quadrature correlations is not excessively larger than one in the initial state, it is still of sufficient magnitude to justify the validity of pillars 2 and 3 and explain the dominance of the Gaussian density correlations in the subsequent dynamics. This is explicitly verified in the simulation plots of Fig. 15. We would like to stress that these simulations are based on the classical field approximation using estimates of the (not directly measurable) experimental parameters $J$ and $\beta^{-1}$. In addition to the above, the temporal profile of the observed decay of non-Gaussianity has been shown to be consistent with simple geometric arguments following from the light-cone propagation of hydrodynamic fluctuations, which are more rigorously backed by analytical calculations presented in Appendix B.

Overall, it seems fair to say that the evidence laid out before suggests that the canonical transmutation mechanism is primarily at play in the experiment and constitutes the underlying explanation of the decay of non-Gaussian correlations. The spatial scrambling mechanism also contributes a partial decrease of non-Gaussianity, despite the absence of dynamical delocalization, but that alone is not sufficient to explain the observed decrease quantitatively, without the additional properties giving rise to the canonical transmutation mechanism.

# 6 Conclusions

To conclude, in this work, we have observed the dynamical emergence of Gaussianity from an initially non-Gaussian state and comprehensively discussed the mechanism responsible for it. We explained the experimental observations and findings with a simple model of canonical rotation and mixing of phononic modes and properties of the initial state. Our analysis stresses the importance of the, often overlooked, higher-order correlations in characterizing and identifying not only equilibrium states but also dynamical mechanisms in quantum systems.

Future experiments will aim at further exploiting the potential of controllable space and time dependent cold-atom traps based on digital micromirror devices [56, 57], which is what made the implementation of box shaped traps and the observation of recurrences of non-Gaussianity possible. Given the high levels of control that are technologically reached in the experimental platform considered here, in conjunction with the perspective of reaching regimes beyond the effectively classical description, this set-up can be regarded as a promising *dynamical quantum simulation* [60, 61]. Such dynamical quantum simulations provide new insights into the dynamics of interacting quantum systems, in instances beyond the scope of known classical simulation methods. The understanding of the precise dynamical mechanisms at work in this context is also important for realizing *quantum field machines* [57], an idea that involves ultra-cold continuous atomic systems to perform quantum thermodynamic tasks and to treat them as instances of thermodynamic machines in situations in which quantum effects

are expected to play a role.

It would be interesting to investigate the effect of particle statistics on the two mechanisms studied here. Due to the boson-fermion correspondence in one spatial dimension the effective description of the experimental system can be expressed equivalently in terms of either bosonic or fermionic degrees of freedom and in fact the quasi-fermionic nature of excitations has been demonstrated in a recent experiment [62]. This suggests that, at least the canonical transmutation mechanism, where locality plays no role, should be relevant also in fermionic systems. Even though this mechanism is new and has not been studied before under more general settings, we indeed expect it to be applicable to certain cases of interaction quenches in superconducting matter, spinful fermions, spin ladders, or other systems involving two or more types of fermions that get mixed by the dynamics, but are initially decoupled and not all are interacting. In such cases the non-Gaussianity of initial correlations would oscillate between the different types, similarly to the present problem. This mechanism is expected to be more clearly distinguishable from the spatial scrambling mechanism in the case of critical dynamics and for smeared observables, which depend more on the long-wavelength excitations that evolve following a linear dispersion relation in this case. The spatial scrambling mechanism, on the other hand, has been shown to work equally well for bosonic or fermionic degrees of freedom and spin chains that can be mapped to non-interacting lattice fermions [7], even though the non-local nature of this mapping (Jordan–Wigner transformation) clearly plays a non-trivial role in the validity of the relevant conditions.

Another question for future study is the possible effects of topological excitations (sine-Gordon solitons) on the two Gaussification mechanisms. As discussed earlier, solitons are rare in the initial state ensembles of scans of the present analysis, but earlier experiments [22] have demonstrated significant presence of solitons in states prepared through a fast non-equilibrium process (for example, fast evaporative cooling) within the coupled double well regime. Given that the validity of some of the pillars of the two mechanisms is shown only in equilibrium and can be affected by the presence of initial solitons, it is unclear what to expect in such a case.

On a more conceptual level, the work presented here can be seen as a comprehensive discussion of the mechanisms of the emergence of Gaussian correlations in physical systems, a type of correlations that is ubiquitous in physics, to say the least. Our theoretical study is matched and underpinned by a body of fresh experimental data that exemplify Gaussification in time in a clear-cut fashion. The diagnostic tools that a tomographic recovery offers provide novel insight into the precise mechanism that is at work here. It is our hope that the present work inspires further studies on the emergence of apparent equilibrium in non-equilibrium quantum dynamics, a field of research at the interface of strongly correlated quantum systems and quantum field theory, of quantum information theory, and of statistical physics.

## Acknowledgements

Joint work at FU Berlin and TU Wien has been supported by the DFG (FOR 2724 on 'Thermal machines in the quantum world' and CRC 183) and the FQXi on 'Fueling quantum field machines with information', for which it constitutes important inter-node preparatory theoretical-experimental work in the development of quantum field machines. FUB has also received funding from the European Union's Horizon2020 research and innovation programme under grant agreement No. 817482 (PASQuanS) on programmable quantum simulators. This work touches also upon identifying platforms of programmable cold atomic quantum simulators, as being funded by BMBF (FermiQP), and on methods of verification of quantum simulators, as funded by the Munich Quantum Valley (K8). It has also been supported by the DFG/FWF Collaborative Research Centre 'SFB 1225 (ISOQUANT)' and the ESQ Discovery Grant 'Emergence

of physical laws: From mathematical foundations to applications in many body physics' of the Austrian Academy of Sciences (ÖAW). F. C., F. S. M., B. R., J. Sabino and T. S. acknowledge support by the Austrian Science Fund (FWF) in the framework of the Doctoral School on Complex Quantum Systems (CoQuS). T. S. acknowledges support by the Max Kade Foundation through a postdoctoral fellowship. J. Sabino acknowledges support from Fundação para a Ciência e a Tecnologia (Portugal) through Project No. UIDB/EEA/50008/2020 and from the DP-PMI and FCT (Portugal). S. S. has been also supported by the Slovenian Research Agency (ARRS) under grant QTE (N1-0109) and by the ERC Advanced Grant OMNES (694544). J. E., M. G., J. Schmiedmayer and S. S. thank the Erwin Schrödinger Institute for its hospitality and support under the programme 'Quantum Simulation–from Theory to Application' (LCW 2019).

## A Tomonaga-Luttinger liquid description of the experimental system

The theoretical description of the experimental system is based on bosonization or Tomonaga-Luttinger liquid (TLL) theory. The system consists of a cold atomic gas confined in two parallel one-dimensional traps at short distance. The two components of the gas are coupled to each other with their coupling being controlled by the height of the barrier between the two traps. The system is therefore described by the Hamiltonian

$$
H = \sum_{i=1,2}\Bigg[ \frac{\hbar^2}{2m}\int \mathrm{d}x\, \partial_x \Psi_i^\dagger(x)\partial_x \Psi_i(x) + \int \mathrm{d}x\mathrm{d}x'\, V(x-x')\Psi_i^\dagger(x)\Psi_i(x)\Psi_i^\dagger(x')\Psi_i(x')
$$
$$
+ \int \mathrm{d}x\,(V_{\mathrm{ext}}(x)-\mu)\Psi_i^\dagger(x)\Psi_i(x)\Bigg] - \hbar J \int \mathrm{d}x\,\big(\Psi_1^\dagger(x)\Psi_2(x) + \Psi_2^\dagger(x)\Psi_1(x)\big), \tag{47}
$$

where $\Psi_i(.)$ is a two component boson field, $V(.)$ the inter-particle interaction, $V_{\mathrm{ext}}(.)$ the external trap potential, $\mu$ the chemical potential and $J$ the tunnelling coupling between the two components. The inter-particle interaction is practically point-like $V(x-x') = \frac{1}{2}g\delta(x-x')$ for points $x$ and $x'$. The confining trap is in general inhomogeneous in the longitudinal direction (parabolic $V_{\mathrm{ext}}(x)$), although homogeneous box traps have also been used in the experiment. In both cases the edges of the system $x = \pm L/2$ are characterized by vanishing particle current at all times. Let us assume for the moment that the trap is homogeneous.

In the *bosonization* or TLL description [63], each of the bosonic fields is expressed in terms of density and phase

$$
\hat{\Psi}^\dagger(x) = \sqrt{\hat{\rho}(x)}\mathrm{e}^{\mathrm{i}\hat{\phi}(x)} \tag{48}
$$

and the density $\rho_i(.)$ is represented as

$$
\hat{\rho}(x) = \left(n(x) - \frac{1}{\pi}\partial_x\hat{\theta}\right)\sum_{\ell=-\infty}^{+\infty} \exp[2\ell\pi\mathrm{i}(\int^x n(x')\mathrm{d}x' - \frac{1}{\pi}\hat{\theta}(x))],
$$

where $n(.)$ is the average density. The auxiliary field $\hat{\theta}$ expresses local deviations of the density from the average value, with $\partial_x\hat{\theta}$ corresponding to long-wavelength density fluctuations and $\exp[2\ell\pi\mathrm{i}(\int^x n(x')\mathrm{d}x' - \frac{1}{\pi}\hat{\theta}(x))]$ corresponding to short-wavelength kinks at the positions of the particles [47, 63]. As long as we are interested in long-wavelength density fluctuations only, we can write

$$
\delta\hat{\rho}(x) := \hat{\rho}(x) - n(x) \approx -\frac{1}{\pi}\partial_x\hat{\theta}\,. \tag{49}
$$

However, it should be kept in mind that the short-wavelength kinks are also low-energy excitations like the long-wavelength fluctuations and can play a significant role in the dynamics. From the bosonic commutation relations of $\hat{\Psi}_i^\dagger(.)$, it can be shown that $\phi, \theta$ obey the canonical commutation relations

$$[\delta\hat{\rho}_i(x), \hat{\phi}_j(y)] = i\delta_{i,j}\delta(x-y). \tag{50}$$

In the same long-wavelength approximation, the particle current $\hat{j}(.)$ is given by

$$\hat{j}(x) := -i\hbar\left(\hat{\Psi}^\dagger(x)\partial_x\hat{\Psi}(x) - \partial_x\hat{\Psi}^\dagger(x)\hat{\Psi}(x)\right) \approx \hbar n(x)\partial_x\hat{\phi}(x), \tag{51}$$

from which we can recognize $\hbar\partial_x\hat{\phi}(.)$ as playing the role of the local velocity field. The long-wavelength fields $\delta\hat{\rho} \sim \partial_x\hat{\theta}$ and $\hat{j} \sim \partial_x\hat{\phi}$ representing the particle density and current respectively are the two fundamental local fields in the hydrodynamic description of the quantum liquid.

From now on we focus on the homogeneous case where the mean density $n$ is assumed to be constant in space (and also in time). Replacing the bosonic field $\Psi$ in the Hamiltonian (47) using the above representation, expanding in powers of $\phi, \delta\rho$ and keeping only quadratic and lowest gradient terms, we obtain the Hamiltonian

$$H = H_1 + H_2 - \hbar J\sqrt{n_1 n_2}\int dx \cos(\hat{\phi}_2(x) - \hat{\phi}_1(x)), \tag{52}$$

where $H_1, H_2$ are of the form of the TLL Hamiltonian

$$H_{\text{TLL}} = \int dx\left[\frac{\hbar^2}{2m}n\left(\partial_x\hat{\phi}_i(x)\right)^2 + \frac{1}{2}g\delta\hat{\rho}_i(x)^2\right]. \tag{53}$$

The Hamiltonian (52) provides a low-energy description of the system. Since the system is symmetric under interchange of the two components, introducing the symmetric $\phi_s = \phi_1 + \phi_2, \delta\rho_s = \frac{1}{2}(\delta\rho_1 + \delta\rho_2)$ and anti-symmetric fields $\varphi = \phi_2 - \phi_1, \delta\varrho = \frac{1}{2}(\delta\rho_2 - \delta\rho_1)$, the Hamiltonian decouples into independent symmetric and anti-symmetric parts $H = H_s + H_a$. The symmetric part $H_s$ is a TLL Hamiltonian for the symmetric fields $\hat{\phi}_s$ and $\delta\hat{\rho}_s$

$$H_{\text{TLL}} = \int dx\left[\frac{\hbar^2}{4m}n\left(\partial_x\hat{\phi}_s(x)\right)^2 + g\delta\hat{\rho}_s(x)^2\right], \tag{54}$$

while the anti-symmetric part is

$$H_{\text{sG}} = \int dx\left[\frac{\hbar^2}{4m}n(\partial_x\hat{\varphi}(x))^2 + g\delta\hat{\varrho}(x)^2\right] - 2\hbar Jn\int dx \cos\hat{\varphi}(x), \tag{55}$$

where $\hat{\varphi}$ and $\delta\hat{\varrho}$ are the antisymmetric or simply the relative phase and density fluctuation fields. The expression in the last equation is the sine-Gordon Hamiltonian. We see that the coupling between the two components of the gas plays the role of a Josephson junction corresponding to a cosine self-interaction of the relative phase field.

From now on, our focus will be on the relative density and phase fields described by Eq. (55). Due to the presence of interaction, the ground and thermal states of this model are non-Gaussian in terms of the canonical fields $\hat{\varphi}(.), \delta\hat{\varrho}(.)$. By setting the barrier height to a large value, the two components are decoupled, i.e., $J \to 0$ and the sine-Gordon reduces to

$$H_{\text{TLL}} = \int dx\left[\frac{\hbar^2}{4m}n(\partial_x\hat{\varphi}(x))^2 + g\delta\hat{\varrho}(x)^2\right], \tag{56}$$

which is of the standard TLL form

$$\hat{H}_{\text{TLL}} = \frac{\hbar c}{2} \int dx \left( \frac{\pi}{K} (\delta \hat{\varrho}(x))^2 + \frac{K}{\pi} (\partial_x \hat{\varphi}(x))^2 \right), \tag{57}$$

where

$$c := \sqrt{\frac{gn}{m}} \tag{58}$$

is the *speed of sound* and

$$K := \frac{\hbar \pi}{2} \sqrt{\frac{n}{mg}} \tag{59}$$

is the *Luttinger parameter*. In the opposite limit of small barrier height, the two components are strongly coupled, i.e., $J$ is large, and the equilibrium properties are determined by the parabolic approximation of the cosine interaction, that is, the KG Hamiltonian

$$H_{\text{KG}} = \int dx \left[ \frac{\hbar^2}{4m} n (\partial_x \hat{\varphi}(x))^2 + g \delta \hat{\varrho}(x)^2 \right] + \hbar J n \int dx \, \hat{\varphi}^2(x). \tag{60}$$

The phonon excitations are now massive with a mass proportional to $\sqrt{J}$.

# B Tomonaga-Luttinger liquid dynamics of correlations in a homogeneous infinite system

In this section, we are going to calculate the large time asymptotics of phase correlations in a thermodynamically large system based on the assumptions of *i)* Tomonaga-Luttinger liquid dynamics, *ii)* initial clustering of correlations, *iii)* homogeneity of the system and *iv)* the validity of classical field approximation (CFA) for the initial state which as discussed in subsections 3.2 and 3.2 means that the density fluctuation operator has vanishing connected correlation functions and vanishing correlations with the phase operator.

Let us begin by expressing the time evolved phase correlations in terms of initial correlations using the Heisenberg picture and the fact that TLL dynamics is Gaussian. Specifically, the Hamiltonian describing the dynamics is assumed to be (56), therefore the Heisenberg equation of motion for the phase field is the wave equation

$$\partial_t^2 \hat{\varphi}(x,t) - c^2 \partial_x^2 \hat{\varphi}(x,t) = 0, \tag{61}$$

with the general solution in infinite space and for arbitrary initial conditions $\hat{\varphi}(x) = \hat{\varphi}(x,0)$, $\delta \hat{\varrho}(x) = \delta \hat{\varrho}(x,0)$, given by the D'Alembert solution

$$\hat{\varphi}(x,t) = \frac{1}{2} (\hat{\varphi}(x+ct) + \hat{\varphi}(x-ct)) + \frac{\pi}{2K} \int_{x-ct}^{x+ct} \delta \hat{\varrho}(x') \, dx', \tag{62}$$

where we have used $\partial_t \hat{\varphi}|_{t=0} = (c\pi/K) \delta \hat{\varrho}$. As already mentioned, given that the phase field is only measurable in the form of differences between two points, we will need to consider either the non-local phase difference field with respect to some reference point or the local phase derivative field. Let us focus first on the latter from which we can derive the former by integration. From the last relation, we see that the time evolution of $\hat{u} = \partial_x \hat{\varphi}$ is given by

$$\hat{u}(x,t) = \partial_x \hat{\varphi}(x,t) = \frac{1}{2} (\hat{u}(x+ct) + \hat{u}(x-ct)) + \frac{\pi}{2K} (\delta \hat{\varrho}(x+ct) - \delta \hat{\varrho}(x-ct)). \tag{63}$$

The time evolution of $\hat{u} = \partial_x \hat{\varphi}$ can be characteristically expressed as the sum of *left-* and *right-moving* fields defined as

$$\hat{u}(x,t) = \hat{\psi}_+(x,t) + \hat{\psi}_-(x,t), \tag{64}$$

where

$$\hat{\psi}_\pm(x,t) := \frac{1}{2}\left(\hat{u}(x,t) \mp \frac{\pi}{K}\delta\hat{\varrho}(x,t)\right). \tag{65}$$

Their name originates from the observation that they follow the simple time evolution $\hat{\psi}_\pm(x,t) = \hat{\psi}_\pm(x \pm ct, 0)$. From Eq. (63), it is easy to derive time evolved phase correlation functions of any order $\left\langle \prod_{i=1}^n \hat{u}(x_i, t) \right\rangle$ from initial ones. All we have to do is substitute (63) and expand to express the result as a sum of initial velocity and density correlations. The same applies to connected correlation functions, since they are multi-linear with respect to the fields. In this way we see that the time evolution of correlations between a set of points is nothing but the result of mixing initial correlations originating from the past light-cone projections of these points.

This expansion simplifies significantly when we take into account the classical field approximation and the property of clustering of correlations which constrain the initial state. Below we state more formally the precise definition that we need for our argument.

**Definition 1** (Classical field approximation (CFA)). *We say that a state satisfies the CFA property, if*

$$\left\langle \prod_{j=1}^n \delta\hat{\varrho}(x_j) \right\rangle_{\text{con}} = 0, \quad \text{for } n > 2 \tag{66}$$

*and*

$$\left\langle \prod_{i=1}^{n_1} \hat{u}(x_i) \prod_{j=1}^{n_2} \delta\hat{\varrho}(x_j) \right\rangle_{\text{con}} = 0, \quad \text{for all } n_1, n_2 > 0. \tag{67}$$

**Definition 2** (Clustering of correlations). *We say that a state has exponentially clustering correlations if*

$$\left\langle \prod_{i=1}^{n_1} \hat{u}(x_i) \prod_{j=1}^{n_2} \delta\hat{\varrho}(y_j) \right\rangle_{\text{con}} = O(e^{-\sum_{i<j}|x_i-x_j|/\xi - \sum_{i<j}|y_i-y_j|/\xi}), \quad \text{for } n_1, n_2 > 0, \tag{68}$$

*where $\xi > 0$ is a correlation length. If the bound is a polynomially decaying function then we speak of polynomially clustering correlations.*

Let us consider a higher-order connected correlation function with $n > 2$ and unequal positions $x_i \neq x_j$ unless $i = j$

$$\left\langle \prod_{i=1}^n \hat{u}(x_i, t) \right\rangle_{\text{con}} = \frac{1}{2^n} \sum_{\substack{\sigma \in \{-1,1\}^{\times n} \\ \mu \in \{0,1\}^{\times n}}} \left\langle [\hat{u}(x_1 + \sigma_1 ct)]^{\mu_1} [\frac{\pi\sigma_1}{K}\delta\hat{\varrho}(x_1 + \sigma_1 ct)]^{1-\mu_1} \cdots \right.$$
$$\left. \times [\hat{u}(x_n + \sigma_n ct)]^{\mu_n} [\frac{\pi\sigma_n}{K}\delta\hat{\varrho}(x_n + \sigma_n ct)]^{1-\mu_n} \right\rangle_{\text{con}}. \tag{69}$$

Notice that this formula is already simpler than the general case of Gaussian dynamics (11) where the Green's functions characterizing the propagation are not simply localized ($\delta$-like)

functions. This property is directly related to the fact that under TLL dynamics wave-packets do not spread. For $n = 2$, we have

$$\langle \hat{u}(x_1, t)\hat{u}(x_2, t) \rangle = \frac{1}{4} \sum_{\sigma_1, \sigma_2 = \pm} \Bigg[ \langle \hat{u}(x_1 + \sigma_1 ct)\hat{u}(x_2 + \sigma_2 ct) \rangle$$
$$+ \left(\frac{\pi}{K}\right)^2 \langle \delta\hat{\varrho}(x_1 + \sigma_1 ct)\delta\hat{\varrho}(x_2 + \sigma_2 ct) \rangle \Bigg]. \tag{70}$$

Here, we see explicitly that whenever $\sigma_1 = \sigma_2 = \sigma$ then the time dependent correlation function is just a rigid translation of the initial one between the points $x_1 + \sigma ct$ and $x_2 + \sigma ct$.

The field operators in the correlation functions can be freely reshuffled for $n > 2$ because any field commutators resulting from this reordering do not contribute to the connected correlations. After ordering the operators, we use the CFA property to evaluate the terms involving only $\hat{\varphi}(.)$ fields or both $cf(.)$ and $\hat{\varphi}(.)$ fields, which simply vanish since $n > 2$ and we are left with the terms involving $\hat{u}(.)$ fields only

$$\left\langle \prod_{i=1}^{n} \hat{u}(x_i, t) \right\rangle_{\mathrm{con}} = \frac{1}{2^n} \sum_{\substack{\sigma \in \{-1,1\}^{\times n} \\ \mu \in \{0,1\}^{\times n}}} \left\langle \prod_{i=1}^{n} [\hat{u}(x_i + \sigma_i ct)]^{\mu_i} \prod_{i=1}^{n} [\frac{\sigma_i \pi}{K} \delta\hat{\varrho}(x_i + \sigma_i ct)]^{1-\mu_i} \right\rangle_{\mathrm{con}}$$

$$= \frac{1}{2^n} \sum_{\sigma \in \{-1,1\}^{\times n}} \langle \hat{u}(x_1 + \sigma_1 ct) \ldots \hat{u}(x_n + \sigma_n ct) \rangle_{\mathrm{con}}. \tag{71}$$

Further simplification applies when we consider the large time limit of correlations and exploit the clustering of initial correlations of local fields. In this limit the initial correlations involved in the above expansions are non-trivial only if all $\sigma_i$ have the same sign. This is because the fields with $\sigma_i > 0$ propagate to the right and those with $\sigma_i < 0$ to the left and therefore become quickly uncorrelated since they originate from distant points. For sufficiently long times, the light-cone separation of these points $ct$ is substantially longer than the constant diameter of the initial points $\mathrm{diam}\{x_i\}$ and the correlation length $\xi > 0$. We can hence bound asymptotically the connected function as

$$\lim_{t \to \infty} \left\langle \prod_{i=1}^{n} \hat{u}(x_i + \sigma_i ct) \right\rangle_{\mathrm{con}} = \lim_{t \to \infty} O(e^{-ct/\xi}) = 0, \quad \text{for signs } \sigma_i \text{ not all equal.} \tag{72}$$

Therefore we can simplify the connected correlation functions accordingly

$$\left\langle \prod_{i=1}^{n} \hat{u}(x_i, t) \right\rangle_{\mathrm{con}} = \frac{1}{2^n} \left\langle \prod_{i=1}^{n} \hat{u}(x_i + ct) \right\rangle_{\mathrm{con}} + \frac{1}{2^n} \left\langle \prod_{i=1}^{n} \hat{u}(x_i - ct) \right\rangle_{\mathrm{con}}. \tag{73}$$

This means that for late times, what is left are correlations between points at fixed distances. Assuming also translational invariance of the initial state, the two terms above are equal

$$\lim_{t \to \infty} \left\langle \prod_{i=1}^{n} \hat{u}(x_i + ct) \right\rangle_{\mathrm{con}} = \lim_{t \to \infty} \left\langle \prod_{i=1}^{n} \hat{u}(x_i - ct) \right\rangle_{\mathrm{con}} = \left\langle \prod_{i=1}^{n} \hat{u}(x_i) \right\rangle_{\mathrm{con}}. \tag{74}$$

This allows us to drop the dependence on signs and we finally obtain the following result for the large time correlations under the above assumptions

$$\left\langle \prod_{i=1}^{n} \hat{u}(x_i, t) \right\rangle_{\mathrm{con}} = \frac{1}{2^{n-1}} \left\langle \prod_{i=1}^{n} \hat{u}(x_i) \right\rangle_{\mathrm{con}} + O(e^{-ct/\xi}), \tag{75}$$

for $n > 2$ and for $n = 2$

$$\lim_{t \to \infty} \langle \hat{u}(x_1, t) \hat{u}(x_2, t) \rangle = \frac{1}{2} \left[ \langle \hat{u}(x_1) \hat{u}(x_2) \rangle + \left( \frac{\pi}{K} \right)^2 \langle \delta \hat{\varrho}(x_1) \delta \hat{\varrho}(x_2) \rangle \right].$$

In particular, for an auto-correlation function, we have

$$\langle \hat{u}(x, t)^n \rangle_{\text{con}} = \frac{1}{2^{n-1}} \langle \hat{u}(x, 0)^n \rangle_{\text{con}} + O(e^{-ct/\xi}). \tag{76}$$

In a similar way, we can derive the scaling of phase difference correlations which turn out to decrease linearly with time until they reach a saturation value. This is due to a purely

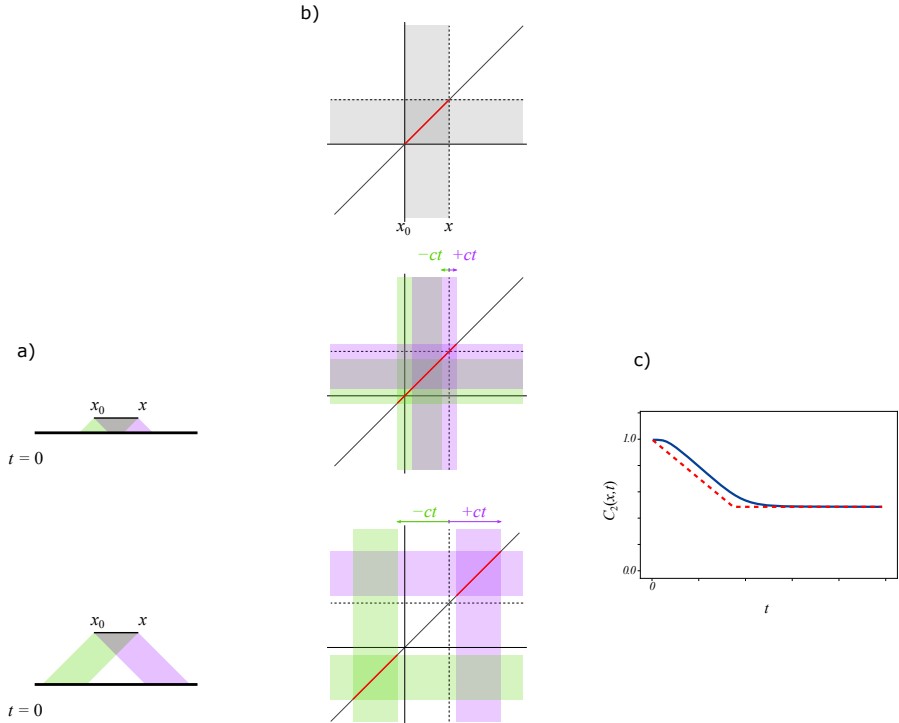

Figure 16: Geometric explanation of the temporal profile of cumulants of the phase difference field $\Delta \hat{\varphi} = \hat{\varphi}(x, t) - \hat{\varphi}(x_0, t)$. (a) Seen from a space-time perspective, the splitting of $\Delta \hat{\varphi}$ into two components (left and right moving) can be depicted by two stripes of different color. For times $t < |x - x_0|/2c$ the two projections on the $t = 0$ line still overlap with each other, while for $t > |x - x_0|/2c$ they do not. (b) To compute the 2nd order cumulant of $C_2(x, t) = \left\langle \Delta \hat{\varphi}(x, t)^2 \right\rangle_{\text{con}}$ we integrate $\langle \hat{u}(y_1, t) \hat{u}(y_2, t) \rangle_{\text{con}}$ over the variables $y_1$ and $y_2$ in the overlapping region of the corresponding stripes which are shifted from the original interval $[x_0, x]$ by $\pm ct$. Given that this correlation function is of short range, i.e., decays quickly away from the diagonal $y_1 = y_2$, the integration gives simply a value proportional to the length of the intersections of these regions and the diagonal (red lines). The stripes move apart with speed $2c$, so the overall length of the intersection intervals at first decreases linearly with time and eventually stops changing. (c) The resulting temporal profile of $C_2(x, t)$ exhibits a linear decrease to a plateau value, which is half of the initial one. The exponential tails of the correlation function away from the diagonal have the effect of smearing the otherwise sharp temporal profile [59]. Following the same arguments applied to higher order cumulants $C_n(x, t)$ we get a decrease by a factor $1/2^{n-1}$.

geometric reason, as we can see by integrating the phase derivative correlations. From (73) the integration interval $[x_0, x]$ is split into two intervals $[x_0 - ct, x - ct]$ and $[x_0 + ct, x + ct]$ corresponding to the light-cone projections when we trace the time-evolved correlations to initial correlations. The two intervals overlap in the sub-interval $[x_0 + ct, x - ct]$ whose length is linearly decreasing with time, up to the time $t_* = |x_0 - x|/(2c)$ beyond which they do not overlap anymore. Since the initial correlations of the phase derivative are short-range, the integral is simply proportional to the length of the overlap, as illustrated in Fig. 14 and in more detail in Fig. 16, resulting in the above described behavior. The large time asymptotic value can be easily related to the initial one

$$
\begin{aligned}
\langle \Delta \hat{\varphi}(x,t)^n \rangle_{\text{con}} &= \int_{[x_0,x]^n} \prod_i^n dy_i \left\langle \prod_i^n \hat{u}(y_i, t) \right\rangle_{\text{con}} \\
&= \frac{1}{2^n} \int_{([x_0-ct,x-ct] \cup [x_0+ct,x+ct])^n} \prod_i^n dy_i \left\langle \prod_i^n \hat{u}(y_i, 0) \right\rangle_{\text{con}} \\
&\rightarrow \frac{1}{2^{n-1}} \langle \Delta \hat{\varphi}(x,0)^n \rangle_{\text{con}} .
\end{aligned}
\tag{77}
$$

For the second cumulant the decrease is by a factor $1/2$, for the fourth cumulant by $1/8$ and so on. As discussed in Subsection 5.2, the above analysis can explain the qualitative characteristics of the temporal profile of the non-Gaussianity measure $M^{(4)}$ in the experiment.

## C Deviations from the Tomonaga-Luttinger liquid dynamics

The TLL Hamiltonian provides a very good description of equilibrium properties of cold atom systems as can be shown by means of *renormalization group* theory [47]. However, it is less clear how accurate this description is in far from equilibrium dynamics, like after a quantum quench. As discussed in the main text, deviations from the TLL model that are irrelevant at equilibrium in the renormalization group sense may be important out of equilibrium. One class of deviations comes from the fact that, in passing from the original Hamiltonian (47) to (54), we ignored powers and gradients of the density-phase fields of order higher than two. Higher power terms in particular induce an effective self-interaction of the phonons resulting in non-Gaussian dynamics. Moreover, some of these corrections correspond to coupling between the symmetric and anti-symmetric modes, meaning that the dynamics of the anti-symmetric modes is not completely closed. In Sec. 4.3 of the main text we have discussed the dynamical effects of deviations from the TLL description and presented evidence that such deviations are negligible in the experiment. As we argued, despite its simplicity the Hamiltonian (54) actually captures the dynamics of the system sufficiently well within the time scales of the experiment.

Here, we will analyze in more detail two of the main deviations from this model that are potentially relevant for the Gaussification mechanism. The first one is related to the presence of inhomogeneity in the system. In general cold atom gases are inhomogeneous due to the longitudinal trapping potential, which is typically parabolic. However, in the present experiment a homogeneous box trap with hard walls was used, so that the atom density was practically homogeneous. It is still interesting to observe that even in the parabolic trap case, the time evolution does not induce delocalization as required for the spatial scrambling mechanism. The second type of deviation refers to non-linear corrections to the phonon dispersion relation, which are present due to the higher gradient terms that were ignored in the derivation of (54). These are quadratic in the density and phase fields and therefore preserve the Gaussianity of dynamics but induce dispersive spreading of the phononic excitations. Since

these physical effects are relevant in Gaussification, it is important to estimate their role in the experiment.

## C.1 Effects of a non-uniform trap potential

The trap inhomogeneity can be taken into account by allowing the density profile $n$ in (54) to be a function of the spatial coordinate $x$

$$\hat{H}_{\text{inhom}} = \int dx \left[ \frac{\hbar^2 n(x)}{4m} (\partial_x \hat{\varphi}(x))^2 + g \delta \hat{\varrho}(x)^2 \right]. \tag{78}$$

In the experimental settings, the Luttinger parameter $K$ is sufficiently large in which case the *Thomas-Fermi approximation* is applicable. In this approximation and for a parabolic trap $V_{\text{ext}}(x) = \frac{1}{2}\omega^2 x^2$, the density profile $n(x)$ turns out to be

$$n(x) = n_0 \left( 1 - \frac{x^2}{R^2} \right) \Theta(R^2 - x^2), \tag{79}$$

where $n_0 = \frac{1}{2}\omega^2 R^2$ is the density at the middle and $R = \sqrt{2\mu}/\omega$ is the semi-classical (or Thomas-Fermi) radius of the system, which is half of the system size.

The Hamiltonian (78) corresponds to an inhomogeneous TLL [64]. In the present special case (weak boson interaction and parabolic trap) an exact solution is possible by expanding the fields $\hat{\varphi}$ and $\delta \hat{\varrho}$ in Legendre polynomials instead of the cosine plane waves of the homogeneous case. The energies of these modes are $E_n = \sqrt{n(n+1)}/R$ (instead of integer multiples of $2\pi/L$). Approximate recurrences are expected to occur at integer multiples of $T = 2\pi R$ (instead of $T = 2L$), since at higher mode numbers all energies are close to integer multiples of $1/R$. The inhomogeneity of the trap affects significantly the dynamics. A wave packet initially localized at the centre propagates to the edges following curved light-cones at a position-dependent speed, as expected for a curved background. In addition, however, edge effects are present, which grow and spread inwards to the centre of the trap. Instead of (62), the time evolution of the phase field is now given by

$$\hat{\varphi}(x,t) = \int dy \, G_{\phi\phi}(x,y,t) \hat{\varphi}(y,0) + \int dy \, G_{\phi\rho}(x,y,t) \delta \hat{\varrho}(y,0), \tag{80}$$

where the qualitative form of the phase-phase and density-phase propagators $G_{\phi\phi}(x,y,t)$ and $G_{\phi\rho}(x,y,t)$ in the parabolic trap is presented in Fig. 19, which should be compared with Figs. 17 and 18 for the hard-wall box trap with Neumann or Dirichlet boundary conditions, respectively.

From our analysis we conclude that despite its special characteristics, the dynamics does not induce delocalization that could justify Gaussification via wave packet spreading. In contrast we find that for all relevant aspects, the dynamics up to the maximum time studied in the experiment are unaffected by edge effects and practically the same as that of a thermodynamically large homogeneous system.

## C.2 Effects of a non-linear phonon dispersion

The linear dispersion of the TLL in its standard form (54) is a rather idealistic property that is far from true in realistic systems. The most important source of nonlinear dispersion comes from the Galilean form of the kinetic energy in the original Hamiltonian (47) [65]. In the density-phase representation this reads

$$\hat{H}_{\text{kin}} = \int dx \left[ \frac{\hbar^2 n}{4m} (\partial_x \hat{\varphi}(x))^2 + \frac{\hbar^2}{4mn} (\partial_x \delta \hat{\varrho}(x))^2 \right], \tag{81}$$

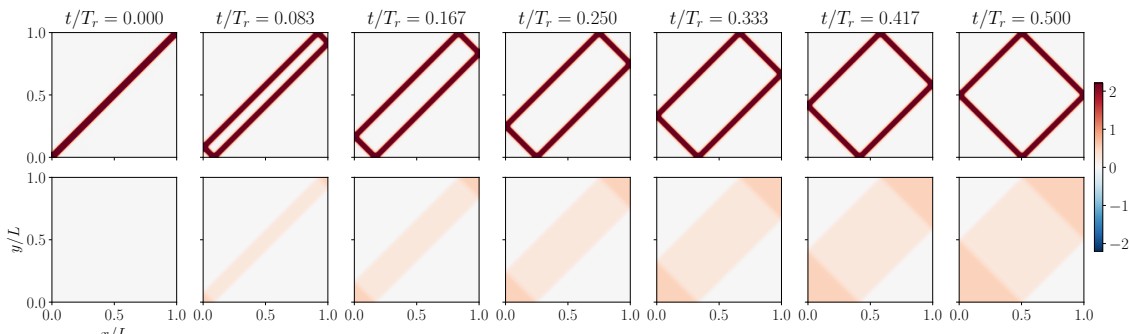

Figure 17: Phase-phase $G_{\phi\phi}(x, y, t)$ (*top*) and density-phase $G_{\phi\rho}(x, y, t)$ propagators (*bottom*) for a homogeneous box with Neumann boundary conditions (real space density plots at various times from zero to half of the recurrence time).

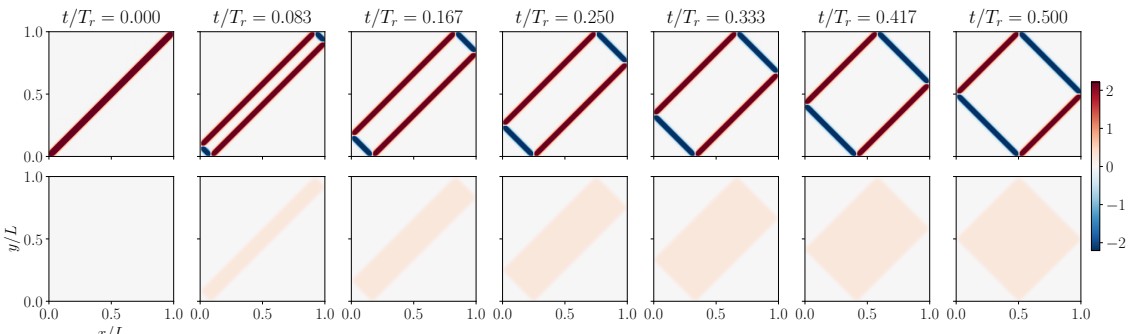

Figure 18: Same as Fig. 17 but for a homogeneous box with Dirichlet boundary conditions.

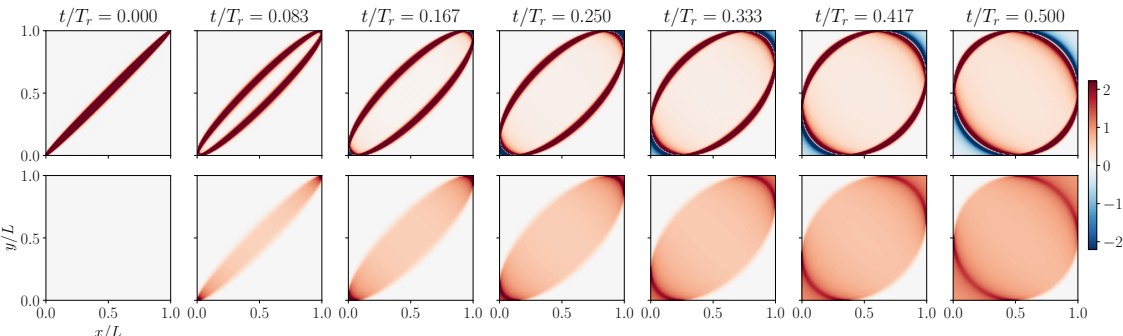

Figure 19: Same as Fig. 17 but for a parabolic trap as described by the Thomas-Fermi approximation.

meaning that the extra term

$$\hat{H}_{\mathrm{disp}} = \frac{\hbar^2}{4mn} \int \mathrm{d}x \, (\partial_x \delta\hat{\varrho}(x))^2 \,, \tag{82}$$

should be added to the TLL Hamiltonian (54). The new Hamiltonian is Gaussian with a different dispersion relation

$$E(k) = \hbar |k| \sqrt{c^2 + \left(\frac{\hbar k}{2m}\right)^2} = \hbar c |k| \sqrt{1 + \left(\frac{\xi_h k}{2}\right)^2} \,, \tag{83}$$

with $c = \sqrt{gn/m}$ as in Eq. (58) and $\xi_{\mathrm{h}} = \hbar/\sqrt{gnm}$ is the *healing length*. This is the Bogoliubov dispersion relation, which is linear at small $k$ ($E(k) \sim \hbar c|k|$ ) consistently with TLL theory, and quadratic at large $k$ ($E(k) \sim \hbar^2 k^2/(2m)$). The linear dispersion of Eq. (54) is an excellent approximation at equilibrium as long as we are interested in length scales much larger than $\xi_{\mathrm{h}}$. In the experiment phase measurements refer to smeared local fields due to the finite imaging (spatial) resolution which is of the order of $32\,\mu\mathrm{m}$. On the other hand, the healing length is estimated to be $\xi_{\mathrm{h}} \approx 0.352\,\mu\mathrm{m}$, much smaller than the smearing length. Therefore for equilibrium states the short distance effects of the nonlinear dispersion (83) are unobservable and negligible. This is not necessarily true, however, for the dynamics: As the non-linearity of the dispersion relation induces spreading of local fields following an algebraic scaling with time, this spreading effect will eventually become significant and manifest itself in the measurements, even if these are restricted to smeared local fields. Nevertheless, by evaluating the spreading effect in the time scale of the experiment, i.e., up to the recurrence time, we find that it is still negligible and unimportant.

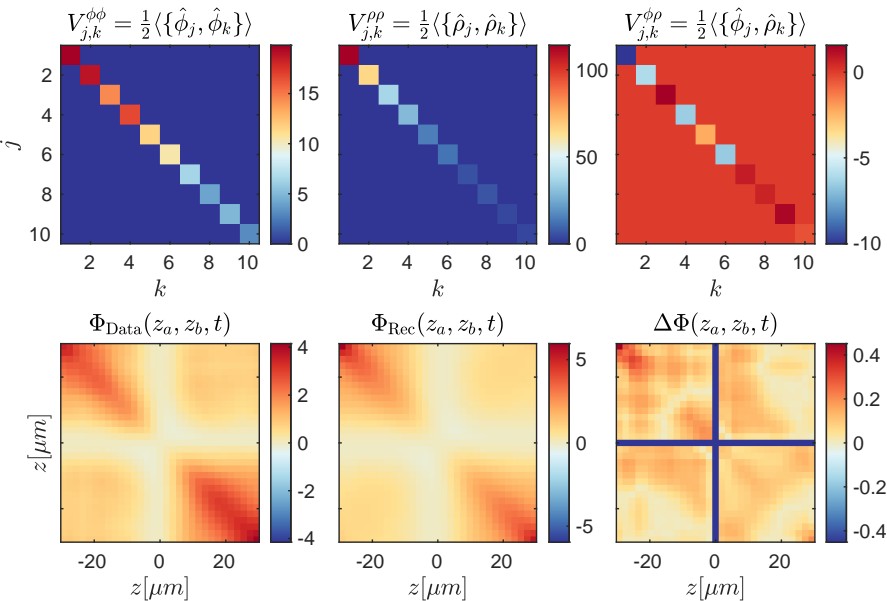

Figure 20: A reconstruction for the same experimental scan as in Fig. 8 but with a restriction of the eigen-mode correlations to the diagonal, i.e., instead of a reconstruction over unrestricted Gaussian states, here, we restrict the variational states with an additional product-state constraint in eigen-mode space. We notice a remarkable stability of the values of correlations on the diagonal. The absence of off-diagonal eigen-mode correlations due to the restriction leads to a reduced fidelity of the fit. While the precision is a little worse, the accuracy of extrapolation of the time evolution is improved as the restriction forces proximity to steady states which behave more stably upon extrapolation outside of the input time window.

# D   Further tomographic reconstruction plots

In this section we include additional plots of tomographic reconstructions of one of the experimental scans (Figs. 20, 21 and 22) that provide more details on the quality of the reconstruction for different choices of the variational states or of the number of modes used in the tomographic method.

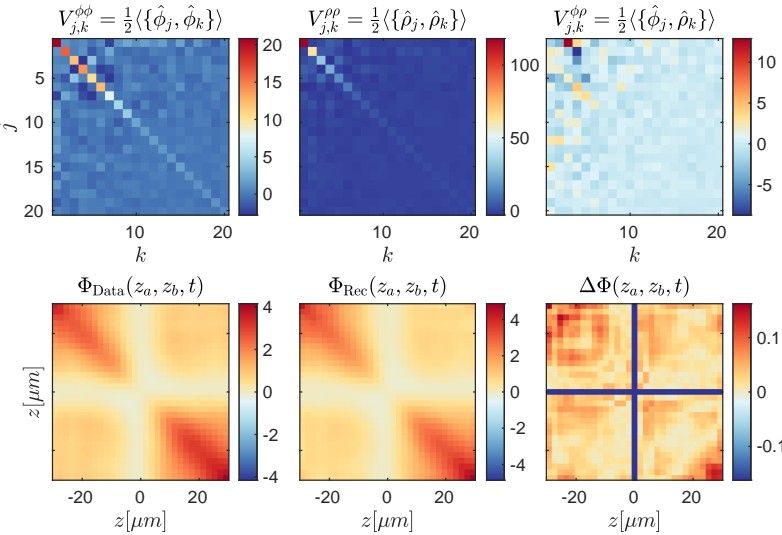

Figure 21: A reconstruction for the same experimental scan as in Fig. 8 but now allowing for an larger number of eigen-modes $\mathcal{N} = 20$. Comparing with the earlier reconstruction using $\mathcal{N} = 10$ eigen-modes we find a rather good convergence in the sense that the correlations which are well-resolved in space and time (modes with $k \leq 7$) are accurately reproduced in both reconstructions. It should be stressed that the time step of the measurements and the spatial resolution of the read-out camera do not allow the accurate resolution of higher momentum modes, and the reconstruction favors diagonal non-squeezed correlations in these modes. The inclusion of additional modes softens the hard cut-off at high momentum modes leading to less off-diagonal artifacts.



Figure 22: The real space representation of the reconstructed density fluctuation covariance matrix for the same experimental scan as in Fig. 8 but based on $\mathcal{N} = 20$ eigen-modes. We see a much more pronounced localization in coordinate space since the inclusion of higher momentum modes allows us to observe more short-range structures. Most notably in comparison to $\mathcal{N} = 10$ we observe the typical anti-correlation stripes close to the diagonal.

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
