# Peer review of "Mechanisms for the emergence of Gaussian correlations"

_SciPost Physics, doi:SciPost Phys. 12, 113 (2022)_

## Round 1 · Referee Report · Anonymous (Referee 1) · 2021-11-9

Strengths

  • timely subject at the centre of the modern cond-mat and stat-phys;
  • very detailed discussion nicely connecting theoretical and experimental point of views;
  • introduces a new theoretical framework (the canonical transmutation) for understanding memory-loss in non-equilibrium dynamics.

Weaknesses

  • just minor things listed in the requested changes.

Report

In this work the authors discuss and discriminate between two mechanism of decay of non-gaussian correlations in non-equilibrium dynamics. They consider a quantum quench setup in which the interactions in the system are switched off abruptly and the proceeding dynamics is of the isolated, non-interacting, system. As observed in the experiments (described by some of the authors in earlier works) the correlations become Gaussian - a process dubbed by the authors "Gaussification". The question is what is the mechanics behind this process. The discussion is based on the experiments with cold-atomic gases employing the atom-chip setup.

This work builds upon the wealth of experimental and theoretical work of the authors and offers a new interpretation of the experimental results. The authors introduce two possible mechanisms of the Gaussification: spatial scrambling and canonical transmutation. Whereas two mechanisms are likely at play in the discussed experiment, it's the latter that the authors claim to be the more important. The canonical transmutation is based on the following (simplified) picture. The correlations of the initial state (as verified in the work) can be divided into Gaussian and non-Gaussian sectors with the Gaussian sector being dominant. Then, the time evolution is understood as a rotation between the two sectors. The resulting correlations are then Gaussian because of the initial dominance of the Gaussian sector. This picture provides also an explanation of the recurrence of non-Gaussianity observed in the experiments.

The paper offers a detailed discussion of various aspects of the two mechanisms, the experimental setup and its theoretical modelling. It provides a convincing justification for the crucial role of the canonical transmutation for explanation of the experimental data. It certainly deserves a publication in the SciPost Physics.

Requested changes

1) pg. 3 authors write: " Such conditions are quite ubiquitously valid in accordance with the demand that an effect broadly at play in the emergence of statistical mechanics in an isolated system should be quite generically applicable." Please rewrite this sentence, the logic seems to be circular here.

2) pg. 4) the authors mention that the spatial scrambling mechanism might not work for nonlinear relations between observables and effective degrees of freedom or when the dispersion relation is linear. It would be beneficial to comment how the two pillars are then broken.

3) typo in "initial" in the caption of Fig. 2.

4) pg. 10 the authors write "However, the entire point why the atom chip experiments are physically illuminating is that, while $\rho(t)$ cannot be acquired as the system involves many-degrees of freedom possibly interacting with each other and so its state is quite complex, the correlation functions can instead be obtained experimentally and be used to characterize the state at different times t." - Isn't the importance of correlation functions a generic situation in the cond-mat setup? What's special about the atom chip experiments in that regard?

5) pg. 10, below eq. (10), how the middle region of the system is defined? Is the trapping potential different for different scans? And if so for which scans is it box-like?

6) Is eq. (7) valid for any $z_i$ or only asymptotically for large relative separations between the points?

7) below eq. (8), what exactly is identified as a non-local Gaussian bath. Please rewrite.

8) Please include definitions of the propagators used in (11) and (12).

9) In the last paragraph of section 2.2 the authors describe with an effective picture the gaussification through canonical transmutation. Could you highlight how exactly this picture arise from the 3 pillars?

10) Fig 5., I don't quite see the linear behavior of the correlation function (mentioned below eq. (22)) for scans 1 and 2 for $r > 10 \mu {\rm m}$. Please comment/rewrite.

11) Below eq. (25) the authors say that for scan 1 , in fig. 6, the large distance asymptote is a constant. Isn't this true also for other scans?

12) The readers would benefit from adding a reference to where eq. (28) were computed.

  • validity: high
  • significance: high
  • originality: high
  • clarity: good
  • formatting: good
  • grammar: good

Author:  Marek Gluza  on 2022-03-09  [id 2275]

(in reply to Report 1 on 2021-11-09)
Category:
answer to question

We would like to thank the referee for the careful and thoughtful report. We are also delighted to read the positive feedback. We have taken the many constructive comments seriously and have accommodated them accordingly in version 2 of our preprint.

**The Referee writes**
>1) pg. 3 authors write: "Such conditions are quite ubiquitously valid in accordance with the demand that an effect broadly at play in the emergence of statistical mechanics in an isolated system should be quite generically applicable." Please rewrite this sentence, the logic seems to be circular here.

**Our answer**
Thank you, we rewrote the sentence as follows:
"Such conditions are quite ubiquitously valid in nature. This is important because statistical mechanics has wide applicability so the prerequisites for its emergence in isolated systems should be broadly fulfilled."

**The Referee writes**
>2) pg. 4) the authors mention that the spatial scrambling mechanism might not work for nonlinear relations between observables and effective degrees of freedom or when the dispersion relation is linear. It would be beneficial to comment how the two pillars are then broken.

**Our answer**
We believe that the referee refers to the next-to-last paragraph of sec.1.1 (p.6 in the first version).
A nonlinear (in particular, non-local) relation between physical (local) observables and the effective degrees of freedom that evolve as non-interacting might mean that the initial clustering condition is not guaranteed for the latter, even if valid for the former. On the other hand, when the dispersion relation of these effective fields is linear (as in the present case) the dynamical delocalisation condition is broken, because, even if they are genuinely local fields, the time evolution does not induce spreading of an initially local wave-packet: the propagating fields travel pinned to the moving light-cone edge points.
We have added this explanation in the relevant paragraph and corrected an inconsistency in the previous paragraph where the initial clustering condition is first introduced: this condition is required to hold for the effective fields that evolve as non-interacting, not just for local observables for which it is natural anyway. It is precisely this distinction why the condition may be broken. We have modified the sentences:
The mechanism of Gaussification by spatial scrambling rests on two essential physical ingredients. Firstly, the initial correlations of local observables must satisfy the condition of clustering, i.e., correlations between observables at distant points must be effectively independent and hence correlations short-ranged.
as:
The mechanism of Gaussification by spatial scrambling rests on two essential physical ingredients. Firstly, the initial correlations of the effective fields in terms of which the dynamics is Gaussian must satisfy the condition of clustering, i.e., their correlations between distant points must be independent and hence their connected correlations must be short-ranged.

and the rest of that paragraph consistently to stress the distinction.

**The Referee writes**
>3) typo in "initial" in the caption of Fig. 2.

**Our answer**
We have now corrected this typographic error.

**The Referee writes**
>4) pg. 10 the authors write "However, the entire point why the atom chip experiments are physically illuminating is that, while ρ(t) cannot be acquired as the system involves many-degrees of freedom possibly interacting with each other and so its state is quite complex, the correlation functions can instead be obtained experimentally and be used to characterize the state at different times t." - Isn't the importance of correlation functions a generic situation in the cond-mat setup? What's special about the atom chip experiments in that regard?

**Our answer**
The referee correctly points out that correlation functions are important in general. In fact it is somewhat surprising that usually researchers in the field of many-body physics only look at second order correlations and completely overlook higher order and connected correlations. The atom-chip experiment of ref. [22] in the paper was the first one that measured these. Apart from offering the possibility to observe a strongly correlated phase of quantum matter (Sine-Gordon model) and being stable enough to perform the measurements, the atom-chip experiment is not special otherwise.

We modified the phrasing in the above sentence by simplifying it to:

When the system involves many-degrees of freedom that interact with each other then its state is quite complex and it is not practical to inquire about its entire density matrix ρ(t). Instead, one should consider correlation functions and indeed these can be obtained experimentally and be used to characterize the system at different times t.

We have also added one more sentence in sec. 6 (Conclusions) to emphasize the significance of higher order correlations in our study:

"Our analysis stresses the importance of higher order correlations, which are often overlooked, in characterizing and identifying not only equilibrium states but also dynamical mechanisms in quantum systems."

**The Referee writes**
>5) pg. 10, below eq. (10), how the middle region of the system is defined? Is the trapping potential different for different scans? And if so for which scans is it box-like?

**Our answer**
Stressing the homogeneity is important because the atoms, also in other platforms, are primarily trapped by a harmonic potential. This leads to inhomogeneities which then have to be corrected by additional engineering. In the case that a non-corrected harmonic trap is used, the middle region of the system is the closest to being homogeneous since the density profile is approximately an inverted parabola and its first derivative vanishes at the midpoint.
However, all scans in the present study correspond to box-like traps, which are a superposition of a harmonic trap with a box trap of much smaller extent so that the density is approximately homogeneous over the entire system, with the exception of the regions very close to the edges, where boundary effects modify the density profile. We added this additional information on p.10 in the paragraph below eq.5 specifying the width of the middle region and of the box-like trap in each scan.

**The Referee writes**
>6) Is eq. (7) valid for any zi or only asymptotically for large relative separations between the points?

**Our answer**
Yes, the clustering property should be understood as a statement for large separations. This is why we do not single out the constants because the fit constants might render the upper bound in eq (7) to be trivial and far off for the short-distance correlations that are influenced by non-universal specifics of the system. On the other hand, for large relative separations it should be expected that the exponential fall-off bound is actually saturated. Reiterating, the equation (7) can be written as is for all non-vanishing distances but might simply be a trivial statement for positions clustered together within the length scale given by the correlation length.

Exponential bounds of this type can be used for spin systems on a lattice without fine-prints but for quantum field theory the positions z_i should not be closer than the momentum cutoff allows because the constant prefactor of the exponential decay will grow with the regularization cutoff to account for the IR singularities that emerge. In our discussion of the experiment this complication doesn’t play a role due to the finite spatial resolution of the measurement. Nevertheless, we have now added explicitly in eq.(7) the condition that it is valid for non-vanishing distances only.

**The Referee writes**
>7) below eq. (8), what exactly is identified as a non-local Gaussian bath. Please rewrite.

**Our answer**
Here, we have tried to identify the Gaussian component of the initial state which is much larger than the non-Gaussian one and plays the role of a Gaussian ‘bath’, analogous to that of the Gaussian canonical sector in the canonical transmutation mechanism. The description ‘non-local’ referred to the fact that this Gaussian component corresponds to long-distance correlations instead of the short-distance non-Gaussian correlations, however in order to avoid confusion due to the different connotation of this word we removed it. We now rewrote this sentence as:
The correlations for points separated beyond the correlation length may intuitively be thought of as playing the role of a Gaussian ‘bath’ in the sense that Gaussification occurs as the result of the dynamical mixing of the initially non-Gaussian component of correlations into the much larger Gaussian component.

**The Referee writes**
>8) Please include definitions of the propagators used in (11) and (12).

**Our answer**
In mathematical terms, the propagators (11) and (12) are the retarded Green’s functions of the initial value problem defined by the Heisenberg equations of motion. Physically, they express the response of the system to a localized disturbance of either of the canonically conjugate fields at the initial time. We have included these definitions in the paragraphs added before and after eq. (11).

**The Referee writes**
>9) In the last paragraph of section 2.2 the authors describe with an effective picture the gaussification through canonical transmutation. Could you highlight how exactly this picture arises from the 3 pillars?

**Our answer**
This effective picture does not stem from the conditions of the canonical transmutation mechanism alone and is only valid in the special case of linear dispersion. For this reason and given that the content of this paragraph (temporal scaling of non-Gaussianity) is discussed in more detail and in the right context later on (sec. 5.2), we now removed the last paragraph from sec. 2.2 and added a sentence about the difference in temporal scaling in the two mechanisms in the last paragraph of sec. 2.3 where it fits:

In the case of linear dispersion systems discussed here, the different nature of the canonical transmutation compared to the spatial scrambling mechanism translates into a different scaling of the approach to equilibrium in a large system, as explained in Subsec. 5.2.

**The Referee writes**
>10) Fig 5. I don't quite see the linear behavior of the correlation function (mentioned below eq. (22)) for scans 1 and 2 for r >10μm. Please comment/rewrite.

**Our answer**
Thank you for pointing out the lack of clarity. We have changed the previous statement

“Inspecting Fig. 5 is indeed what we find, namely for all prepared initial conditions for r >10μm we see a linear behavior for all values of J considered in the experiment.”

to

“Inspecting Fig. 5 we indeed find that for r >10μm a linear behavior is valid, namely for all prepared initial conditions we see either a non-trivial linear increase (scans 3 and 4 corresponding to relatively small $J$) or leveling-off (scan 1 corresponding to the largest $J$ and to a lesser extent scan 2 at an intermediate value of $J$) compatible with a small or essentially negligible slope constant.”

**The Referee writes**
>11) Below eq. (25) the authors say that for scan 1, in fig. 6, the large distance asymptote is a constant. Isn't this true also for other scans?

**Our answer**
The referee correctly points out that in fig. 6 the long-distance asymptote is constant asymptote for all other scans too; in fact the asymptote is zero consistently with our arguments about the decay of velocity field correlations.
What we meant to refer to is fig. 5 instead. Here we see that scan 1 (blue) which corresponds to the KG effective theory has fluctuations that level off towards a constant value while the other scans, especially 3 and 4, are consistent with an increase with separation r. We have modified the text accordingly also taking into account the previous point of the referee.
In principle the same information is contained in fig. 6 but this property is much harder to spot: One would have to integrate finding that the negatively-valued anticorrelation crest around +/- 10 μm approximately cancels the positive auto-correlation peak around 0 μm. Of course given the text statement, fig. 5 should have been referenced and not fig. 6.

**The Referee writes**
>12) The readers would benefit from adding a reference to where eq. (28) were computed.

**Our answer**
We have added a textbook reference for the second part of eq. (28) and further explanation. In particular, we have added further explanation for the choice of the UV regulator (“the contribution of high momentum modes is suppressed by the cutoff Λ, which in the experiment is dominated by the finite imaging resolution which has the form of a Gaussian weight function.”). The first part of eq. (28) is derived straightforwardly taking derivatives with respect to the positions of the two points between which correlations are computed.

Again, we thank the referee for the positive and insightful report. Having carefully accommodated all comments, we hope that this work is now suitable for publication in its present form.

---

## Round 1 · Referee Report · Anonymous (Referee 2) · 2022-1-25

Strengths

1-Very clearly written paper
2-Self-contained in terms of methods and details
3-Nice combination of theory and experiment

Weaknesses

1-Perhaps too long...

Report

This work suggests and analyses two dynamical mechanisms of emerging Gaussification: one by spatial scrambling and the second by canonical transmutation. These two mechanisms are used to interpret experimental results by studying higher-order correlation functions. Discussion is very clear, the paper is self-contained and well structured. I have essentially two questions. It seems that the two mechanisms suggested here are generic, so: 1) what is the effect of statistics (if any) of canonical variables (e.g. bosonic or fermionic) in these mechanisms and what happens in spin systems where canonical variables are not so obvious as in the bosonic case studied in this work? 2) what would be the effect of topological excitations (for example solitons in the sine-Gordon model studied in the paper) in these mechanisms?

Requested changes

One optional change: Addition of discussion related to the two questions in the report.

  • validity: top
  • significance: top
  • originality: top
  • clarity: top
  • formatting: perfect
  • grammar: perfect

Author:  Marek Gluza  on 2022-03-09  [id 2274]

(in reply to Report 2 on 2022-01-25)
Category:
answer to question

We would like to thank the referee for this thoughtful and at the same time positive report. We are delighted to read that the referee thinks that our “discussion is very clear, the paper is self-contained and well structured”.

**Answer to question 1)**
The spatial scrambling mechanism works equally well in fermionic systems and the delocalisation condition is generally satisfied in local lattice models or certain spin chains (ref. [7] in our work). When such systems follow critical dynamics, then the dispersion relation of the long wavelength excitations is linear but the nonlinearity at shorter wavelengths (comparable to the lattice spacing) ultimately results in the delocalisation needed for the mechanism to work. Nevertheless, at short time scales and for smeared observables the effects of this mechanism are negligible, similarly to the present experiment.

The canonical transmutation mechanism, on the other hand, seems to be new and has not been studied in literature before, neither for bosonic nor for fermionic systems. What may be possible is to consider cases including superconducting terms and ramping the superconducting gap. In this case if the hopping sector is Gaussian and the BCS pairing sector is non-Gaussian, then the mixing between these two could lead to dynamics that can be characterized as transmutation. In other words, for bosons, squeezing has been important to observe non-trivial transmutation dynamics and BCS superconductivity is formally the fermionic analog of that.
For Gaussification by canonical transmutation additionally the Gaussian sector ‘pillar’ is necessary. Here one could for example consider the hopping correlations to have a fixed correlation length while the BCS correlations could be paired in momentum, possibly enabling the identification of a natural and singled out Gaussian ‘reservoir’. For more than one type of fermions (e.g. spin-up electrons mixing with spin-down) only one of which is directly measurable and which are coupled to each other by the dynamics, the above mechanism is still applicable. Similarly, this mechanism can be present in the case of e.g. coupled spin ladders (which can be mapped to the previous case), where only one of them is accessible by measurements. There are certain obstructions however that may necessitate a closer look before claiming that the transmutation mechanism leads to Gaussianity, namely, for spins and fermions the local operators are bounded in norm as opposed to bosons. Hence the Gaussian dominance ‘pillar’ may not carry over in exactly the same fashion and may demand additional discussion.
Finally, let us comment on the blurring of the bosonic and fermionic character of quantum particles trapped to one spatial dimension. Recent observations [arxiv:2111.13647] have uncovered that Pauli-blocking is at play in our experimental system, i.e. its effective low-energy behavior stems from blocking high energy excitations with ‘wrong’ rapidities. So while the experiment can be discussed using a bosonic framework it is also worthwhile to understand fermionization of the system even away from the Tonks-Girardeau limit.

**Answer to question 2)**
This is yet another interesting point. Let us first discuss the effect on the transmutation mechanism: Additional solitons originating from a non-equilibrium process (for example fast evaporative cooling) will lead to additional (non-Gaussian) fluctuations of the phase, making the non-Gaussian sector bigger. Whether Gaussification will occur, depends on whether the Gaussian sector still dominates, i.e., on how many additional solitons there are. On the basis of semiclassical arguments, if the solitons have a velocity, they are also expected to contribute to the initial density fluctuations. Whether this contribution is Gaussian or not is unclear to us and probably depends on the specifics of the process generating the solitons.
The effect of the initial presence of solitons on the dynamics can be understood theoretically for the spatial scrambling mechanism. For this, the validity of the two ‘pillars’ would need to be verified. If the non-local nature of solitons does not lead to non-local correlations then together with delocalization one could predict Gaussification by spatial scrambling even in presence of initial solitons. For the experiment at hand, both the ‘pillars’ of this ‘older’ mechanism seem to be unclear, further illustrating why the appearance of Gaussification by canonical transmutation is an interesting effect.

**Implemented changes**
In version 2 of the preprint we have added a discussion of these questions in sec. 6 (Conclusions), fourth and fifth paragraph.

---

## Round 2 · Author Response

Dear Editors,

We would like to thank the referee for their feedback, corrections and suggestions for improvement, which we have thoroughly taken into account in the new version. We note that both referees are satisfied with the quality of our work and the significance of our findings. Consequently, they both ultimately recommend publication in SciPost Physics. In the following, we address their questions and comments in detail and explain what we have done to accommodate the concerns raised.

The comments of the first Referee have pushed us to expand on the mechanism of Gaussification by canonical transmutation, specifically to discuss how the discussed effect may come into play in systems other than the phononic excitations of cold atomic gases which we have focused on. We believe that this resulted in significantly more merit in the outlook and conclusions of our work.

In turn, the second Referee provided several very relevant comments. We took them very seriously and were happy to do so because it allowed us to iron out some of the statements that were important to the crux of our work. We feel glad about having had the opportunity to clarify the passages pointed out by the Referee because these should be much clearer now.

With best regards,
The Authors

---

## Round 2 · List of Changes

List of changes:
- added a paragraph in conclusions about Gaussification by canonical transmutation in systems other than 1d cold atomic gases
- added a paragraph in conclusions discussing topological excitations in context of Gaussification by canonical transmutation
- clarified sentence about conditions for emergence of statistical mechanics
- rewrote a paragraph clarifying aspects of Gaussification by spatial scrambling in effective field theory
- clarified the passage on the role of correlation functions in experiments on quantum many body systems
- added more information about the role of homogeneity in the experiment
- clarified formula (7) excluding overlapping positions
- clarified description related to the role of initial Gaussian correlations
- described the propagators in eqs (11-12)
- corrected the formatting of the caption on p. 8
- improved description of the temporal ongoing of Gaussification by canonical transmutation
- clarified the description of the meaning of fig 5
- corrected the referral to fig 5 and not 6 below eq (25)
- added reference to the formula (28)

---

## Editorial Decision

published